# The C-terminal tail of polycystin-1 suppresses cystic disease in a mitochondrial enzyme-dependent fashion

Laura Onuchic [1], Valeria Padovano [1], Giorgia Schena[1], Vanathy Rajendran [1], Ke Dong[2], Xiaojian Shi[1,3], Raj Pandya [1], Victoria Rai [1], Nikolay P. Gresko[1], Omair Ahmed[1], TuKiet T. Lam [4,5], Weiwei Wang[5], Hongying Shen [1,3], Stefan Somlo [2] & Michael J. Caplan [1] ✉

Autosomal dominant polycystic kidney disease (ADPKD) is the most prevalent potentially lethal monogenic disorder. Mutations in the *PKD1* gene, which encodes polycystin-1 (PC1), account for approximately 78% of cases. PC1 is a large 462-kDa protein that undergoes cleavage in its N and C-terminal domains. C-terminal cleavage produces fragments that translocate to mitochondria. We show that transgenic expression of a protein corresponding to the final 200 amino acid (aa) residues of PC1 in two *Pkd1*-KO orthologous murine models of ADPKD suppresses cystic phenotype and preserves renal function. This suppression depends upon an interaction between the C-terminal tail of PC1 and the mitochondrial enzyme Nicotinamide Nucleotide Transhydrogenase (NNT). This interaction modulates tubular/cyst cell proliferation, the metabolic profile, mitochondrial function, and the redox state. Together, these results suggest that a short fragment of PC1 is sufficient to suppress cystic phenotype and open the door to the exploration of gene therapy strategies for ADPKD.

Autosomal dominant polycystic kidney disease (ADPKD) is the most common life-threatening monogenic disease, with a prevalence of ~1:1000[1]. It is characterized by the progressive development of fluid-filled cysts whose expansion compromise renal function and can lead to end-stage renal disease. Mutations in the *PKD1* gene, which encodes polycystin-1 (PC1), are responsible for ~78% of cases[2]. Although *PKD1* was identified over 25 years ago[3], the downstream pathways leading to cystogenesis have not been fully elucidated, and current therapeutic options remain scarce and incompletely effective.

Over the past decade, metabolic abnormalities have emerged as a hallmark of ADPKD[4]. The first evidence for metabolic alterations in ADPKD came from the observation of increased glycolysis and lactate production in cells from a *Pkd1* knockout (KO) mouse model[5]. Limiting

glucose availability decreased these cells' proliferation, and treatment with 2-deoxyglucose (2DG) to inhibit glycolysis led to partial amelioration of the cystic phenotype in *Pkd1*-KO mice[5,6]. The dependence on glycolysis and the increased lactate levels are together suggestive of an aerobic glycolysis phenotype, similar to the Warburg effect observed in cancer cells[5–7]. Subsequently, other significant metabolic alterations have been observed in ADPKD cellular and animal models, such as defective fatty acid oxidation and decreased rates of oxidative phosphorylation[8–10]. Several promising therapeutic approaches that target these altered metabolic pathways have been explored in animal models and have produced phenotypic benefits, including anti-miR-17 or fenofibrate leading to increased levels of PPARα[9], as well as metformin[11], food restriction[12] and ketogenic diet[13]. The mechanisms

[1]Department of Cellular and Molecular Physiology, Yale University School of Medicine, New Haven, CT 06510, USA. [2]Department of Internal Medicine and Division of Nephrology, Yale University School of Medicine, New Haven, CT 06510, USA. [3]Systems Biology Institute, Yale University, West Haven, CT 06516, USA. [4]Department of Molecular Biophysics and Biochemistry, Yale University, New Haven, CT 06510, USA. [5]Keck Mass Spectrometry & Proteomics Resource, Yale University School of Medicine, New Haven, CT 06511, USA. ✉e-mail: michael.caplan@yale.edu

through which reduced or absent PC1 function produces metabolic alterations, however, have yet to be fully worked out. Recent data suggest the intriguing possibility that direct physical communication between the PC1 protein and components of the mitochondrion may regulate mitochondrial bioenergetics[10].

PC1 is a large 462-kDa membrane glycoprotein that undergoes cleavage at its N- and C-termini[14]. The N-terminal cleavage occurs at the G protein-coupled receptor Proteolytic Site (GPS), giving rise to a 3048-amino acid (aa) N-terminal fragment (NTF) that remains non-covalently attached to the 1254-aa C-terminal fragment (CTF), which comprises PC1's 11-transmembrane domains and its cytoplasmic tail[15]. PC1 C-terminal cleavage generates several shorter PC1-CTF fragments and C-terminal tail fragments (PC1-CTT). An ~100-kDa transmembrane CTF localizes to the endoplasmic reticulum where it may modulate store-operated calcium entry[16]. Several PC1-CTT fragments ranging from 17 to 34-kDa have been reported to translocate to the nucleus and to mitochondria[17–19]. Of note, expression of a protein construct corresponding to the 17-kDa PC1-CTT fragment rescued the fragmented mitochondrial network detected in cells lacking PC1 expression[19]. Similarly, in vitro expression of a PC1-CTT construct corresponding to the final 200 aa of PC1 decreases cellular proliferation and the cross-sectional area of cysts formed by *Pkd1*-KO cells in 3D culture[20]. In the current study, we expressed the 200 aa PC1-CTT sequence in *Pkd1*-KO ADPKD mouse models and characterized the resultant phenotypes.

We show that PC1-CTT expression significantly reduces the severity of cystic disease and that this effect is dependent upon the interaction between the PC1-CTT and the mitochondrial enzyme Nicotinamide Nucleotide Transhydrogenase (NNT). Expression of PC1-CTT in NNT-deficient mice does not produce disease amelioration. Furthermore, we show that PC1-CTT re-expression in the presence of NNT leads to increased mitochondrial mass, altered redox modulation, increased assembly of ATP synthase at a "per mitochondrion" level, and decreased tubular epithelial cell proliferation. Similarly, unbiased metabolomics reveals that the ability of the PC1-CTT to normalize the *Pkd1*-deficient metabolic profile is consistent with PC1-CTT modifying the normal function of NNT. Finally, we show that PC1-CTT re-expression in *Pkd1*-KO mice can increase NNT enzymatic activity, as measured in mouse renal tissue ex vivo, to the level observed in healthy wild-type (WT) controls. This same enzymatic activity assay, performed utilizing an in vitro cell culture system, reveals that the PC1-CTT mitochondrial-targeting sequence (residing between aa residues 4134–4154 of the PC1 protein) is necessary for the modulation of NNT enzymatic activity. Our discovery that expression of a short fragment of the PC1 protein is sufficient to reduce disease severity in mouse models opens the door to exploration of gene therapy strategies for the treatment of ADPKD.

## Results

### Expression of PC1-CTT suppresses cystic phenotype in an orthologous murine model of ADPKD

We generated a BAC construct in which the sequence that encodes the human PC1-CTT (aa 4104–4303) coupled to an N-terminal 2XHA epitope tag[20,21] is preceded by a Flox-Stop sequence and is inserted into the Rosa26 locus (CTT). A transgenic mouse line that incorporated this BAC transgene stably into its germline was crossed with a previously characterized conditional *Pkd1*-KO mouse model of ADPKD (*Pkd1^{fl/fl};Pax8^{rtTA};TetO-Cre*)[22] on the C57BL/6N ("N") background. Doxycycline induction administered from p28-p42 to these second-generation mice (*2HA-PC1-CTT;Pkd1^{fl/fl};Pax8^{rtTA};TetO-Cre*) leads to CTT expression in renal epithelial cells that have undergone Cre-driven disruption of *Pkd1* and therefore lack PC1 expression (Fig. 1A). We generated cohorts of littermates with comparable sex distributions that did or did not carry the CTT BAC transgene on the "N" background (N-*Pkd1*-KO + CTT vs N-*Pkd1*-KO mice) and evaluated them at 16 weeks of age (Fig. 1B–E and Supplementary Fig. 1). The

mice that expressed CTT presented significant reduction of their cystic burden, revealed by a decrease in median kidney-to-body weight ratio (KW/BW) to 4.03% as compared to 15.78% in N-*Pkd1*-KO animals that did not inherit the transgene (Fig. 1B). Consistent with the preservation of their renal size and morphology, CTT-expressing animals exhibited significant reductions in median blood urea nitrogen (BUN) and serum creatinine (4.1 and 3.9-fold, respectively), in comparison to the N-*Pkd1*-KO littermates. Notably, the levels of both of these kidney function markers did not significantly differ between N-*Pkd1*-KO + CTT animals and healthy WT controls (Fig. 1C, D). Quantitative analysis of CTT expression in N-*Pkd1*-KO + CTT mice revealed levels approximately 1.5-fold above those expected for WT animals (Supplementary Fig. 2a, b). This finding indicates that the suppression of cystic phenotype observed in N-*Pkd1*-KO + CTT mice was not a consequence of massive overexpression of CTT.

### PC1-CTT interacts and colocalizes with the mitochondrial enzyme Nicotinamide Nucleotide Transhydrogenase (NNT)

We next employed a BAC transgenic mouse model (*Pkd1^{F/H}*-BAC)[23,24], generated on a mixed strain background[23] (referred to as BAC-*Pkd1*), to identify pathways linking PC1-CTT expression and disease suppression. This mouse line expresses PC1 tagged with a 3XFLAG epitope at its N-terminus and with a 3XHA sequence inserted prior to the stop codon (3FLAG-PC1-3HA) at its C-terminus, under the control of its native promoter. Previous data showed that full-length PC1 localizes to mitochondria-associated endoplasmic reticulum membranes (MAMs)[10], while PC1-CTT localizes to mitochondrial matrix[19]. We sought to identify interaction partners of these mitochondria-associated pools of PC1 and its fragments. A crude mitochondrial fraction from *Pkd1^{F/H}*-BAC mouse kidneys, expected to contain MAM-associated full-length PC1 and PC1-CTT cleavage products, was isolated by differential centrifugation[25]. Following the addition of the cleavable crosslinker DTSSP, mitochondria were solubilized and the lysate was subjected to immunoprecipitation using anti-HA magnetic beads (Fig. 2A). Immunoblot analysis of this precipitate revealed not only the presence of the 150-kDa CTF derived from full-length 3FLAG-PC1-3HA, but also PC1-CTT-3HA fragments of 17–37 kDa (Fig. 2B). We compared mass spectrometric proteomic analyses of *Pkd1^{F/H}*-BAC and WT immunoprecipitates. Potential PC1 and PC1-CTT interactors identified in this way were cross-referenced against the MitoCarta 2.0 mitochondrial proteome[26]. The most significantly enriched interactor identified was Nicotinamide Nucleotide Transhydrogenase (NNT) (Fig. 2C and Supplementary Data 1), a protein that spans the inner mitochondrial membrane (IMM) and connects the proton gradient to the exchange of reducing equivalents between NADH/NAD$^+$ and NADP$^+$/NADPH. Other PC1-relevant hits specific to the endoplasmic reticulum (ER) proteome were identified, consistent with the fact that the crude mitochondria fractions contain MAMs. This is the case for the observed fourfold enrichment of *Ganab* ($P < 0.05$), which encodes the glucosidase II α-subunit. Of note, mutations in *GANAB* are a rare cause of ADPKD[27,28]. Immunoblot analysis of a reverse immunoprecipitation using agarose-conjugated anti-NNT revealed recovery of both the CTF derived from full-length 3FLAG-PC1-3HA and also the PC1-CTT-3HA (Fig. 2D).

We next assessed the subcellular localization and biochemical validity of the putative CTT/NNT interaction. Due to the low expression of CTT in mice[19,24], we could not reliably assess this protein's subcellular localization through immunofluorescence microscopy of kidney tissue sections. Thus, we transiently transfected WT HEK293 cells with the 2HA-PC1-CTT construct and found that it colocalized extensively with endogenous NNT (Fig. 2E, F), which in turn colocalized with TOMM20 (Supplementary Fig. 3) at the mitochondria. Interestingly, the widely used C57BL/6J ("J") mouse strain carries a deletion of exons 7-11 in the *Nnt* gene, a mutation that completely abrogates NNT expression[29] (Fig. 2G). We moved the alleles required to

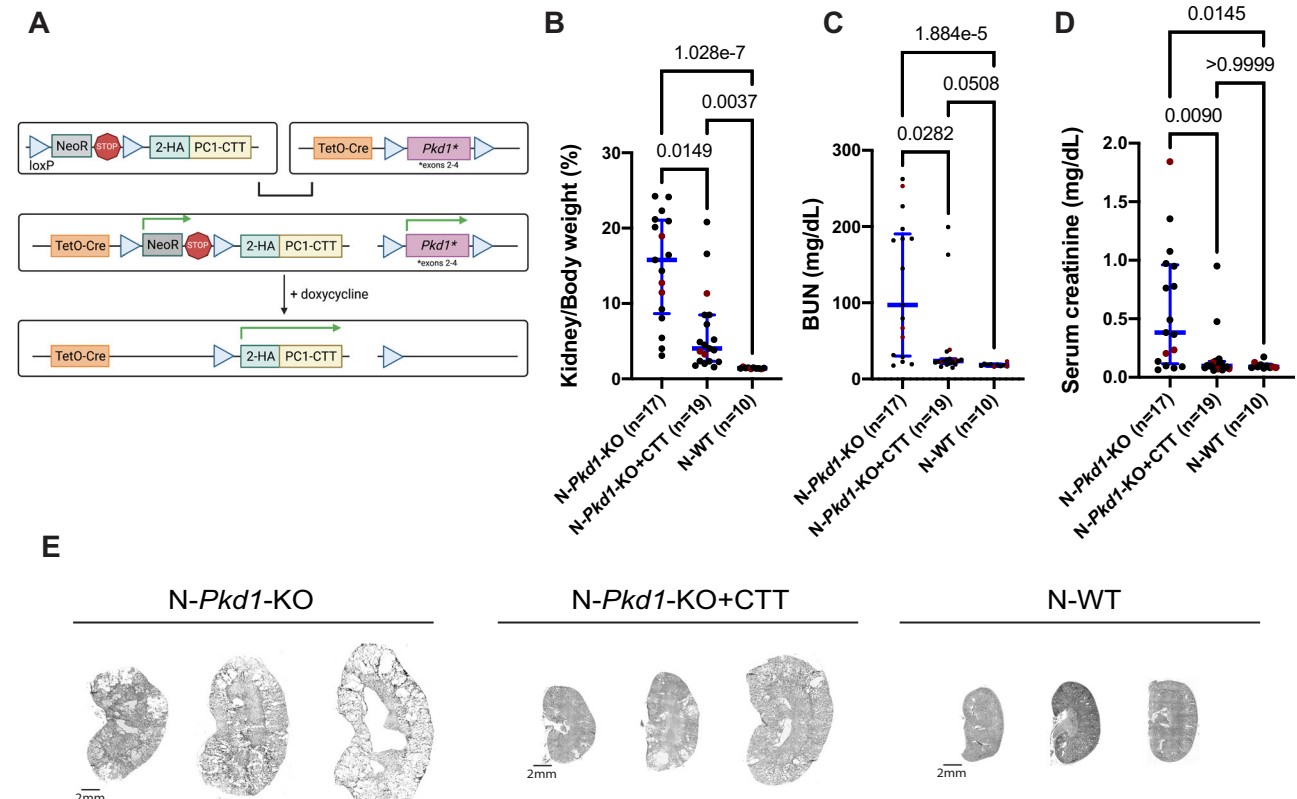

**Fig. 1 | Expression of polycystin-1 C-terminal tail (CTT) suppresses cystic disease in an orthologous mouse model of ADPKD. A** Design of the *2HA-PC1-CTT;Pkd1^fl/fl^;Pax8^rtTA^;TetO-Cre (Pkd1-KO + CTT)* mouse model. We generated transgenic mice that carry a BAC-*2HA-PC1-CTT* transgene inserted in the Rosa26 locus and preceded by a neomycin resistance (NeoR) STOP cassette flanked by *lox*P sequences. These mice were crossed with the previously characterized *Pkd1^fl/fl^;Pax8^rtTA^;TetO-Cre* mouse model of ADPKD in which exons 2–4 of the *Pkd1* gene are flanked by *lox*P sequences[22,74]. Cre-mediated recombination of these second-generation *2HA-PC1-CTT;Pkd1^fl/fl^;Pax8^rtTA^;TetO-Cre* mice via doxycycline induction promotes 2HA-PC1-CTT protein expression and loss of full-length PC1 protein expression in tubular epithelial cells. **B–D** Comparative analysis of N-*Pkd1*-KO, N-*Pkd1*-KO + CTT, and N-WT mice showing differences in KW/BW ratio (**B**), BUN (**C**),

and serum creatinine (**D**). Cystic mouse cohorts are composed of 53–58% female and 42–47% male mice. Red dots represent the animals depicted in (**E**).
**E** Representative H&E-stained kidney sections (4×) from 16-week N-*Pkd1*-KO, N-*Pkd1*-KO + CTT, and N-WT mice. Each group of sections is representative of the average KW/BW ratio of the entire cohort and is illustrative of the inherent variability associated with this mouse model[22,32]. Scale bar: 2 mm. Non-parametric data are depicted with the median and interquartile range (blue bars). Multiple group comparisons were performed using Kruskal–Wallis test followed by Dunn's multiple-comparisons test. H&E-stained kidney sections from all of the cystic mice included in this cohort are provided in Supplementary Fig. 1. Source data are provided as a Source Data file.

---

produce the *Pkd1*-KO ± CTT mice to this NNT-deficient background (J-*Pkd1*-KO and J-*Pkd1*-KO + CTT). Immunohistochemistry confirmed absence of NNT in "J" cystic mice and showed expression of NNT in distal nephron segments and, to a lesser degree, in proximal tubules in both N-*Pkd1*-KO and N-*Pkd1*-KO + CTT mice (Supplementary Fig. 4). This pattern reproduces that reported in both the Human Protein Atlas[30] and in the Rat Kidney Tubule Expression Atlas[31]. Finally, to confirm the CTT/NNT interaction in vivo, we performed anti-HA pull-downs from N-*Pkd1*-KO + CTT, J-*Pkd1*-KO + CTT, and N-*Pkd1*-KO total kidney lysates. We found that NNT is only detected in immunoprecipitates from "N" cystic mice that express CTT and not in those derived from N-*Pkd1*-KO mice that express NNT but not CTT (Fig. 2H). As expected, anti-HA immunoprecipitates from *Pkd1*-KO + CTT kidneys on the "J" background did not contain a 114-kDa NNT band. Taken together, these data demonstrate that PC1-CTT can localize to mitochondria and that it interacts with the IMM protein NNT in mouse kidney epithelial cells in vivo.

### The PC1-CTT/NNT interaction modulates disease progression

We next employed the J-*Pkd1*-KO + CTT mice to evaluate the relevance of the PC1-CTT/NNT interaction to the observed CTT-dependent disease suppression. No significant changes in KW/BW ratio, BUN or serum creatinine levels were observed between J-*Pkd1*-KO + CTT mice

and their J-*Pkd1*-KO littermates (Fig. 3A–E). We further characterized these models in both backgrounds by measuring tubular and cystic area relative to the whole-kidney area. This parameter was significantly smaller in *Pkd1*-KO + CTT vs *Pkd1*-KO mice exclusively on the "N" background (Fig. 3D). Additionally, we performed immuno-fluorescence microscopy on renal tissue from 16-week-old mice and quantitated the fraction of Ki67-positive nuclei as a surrogate assessment of the extent of tubular epithelial cell proliferation. Cyst-lining cell proliferation constitutes a hallmark of cyst expansion in ADPKD[22,32]. While the fraction of Ki67-positive nuclei was reduced by a factor of 2.3 in *Pkd1*-KO + CTT compared to *Pkd1*-KO mice on the "N" background, proliferation levels remained equivalently elevated in these models on the "J" background (Fig. 3F, G).

To ensure that no other factors contribute to the different responses of the "N" and "J" strains to CTT expression, we assessed whether there were any systematic differences between these models in the level of Cre-expression or Cre-induced *Pkd1*-rearrangement. We found that *Pax8^rtTA^* and *TetO-Cre* copy numbers were randomly distributed across the 4 groups and did not correlate with disease severity, as defined by KW/BW ratio in these 16-week animal cohorts (Supplementary Fig. 5a). Similarly, Cre-recombination efficiency, as revealed by levels of non-rearranged *Pkd1* product, was the same across all 4 mouse groups (Supplementary Fig. 5b, c). Of note, the Cre-

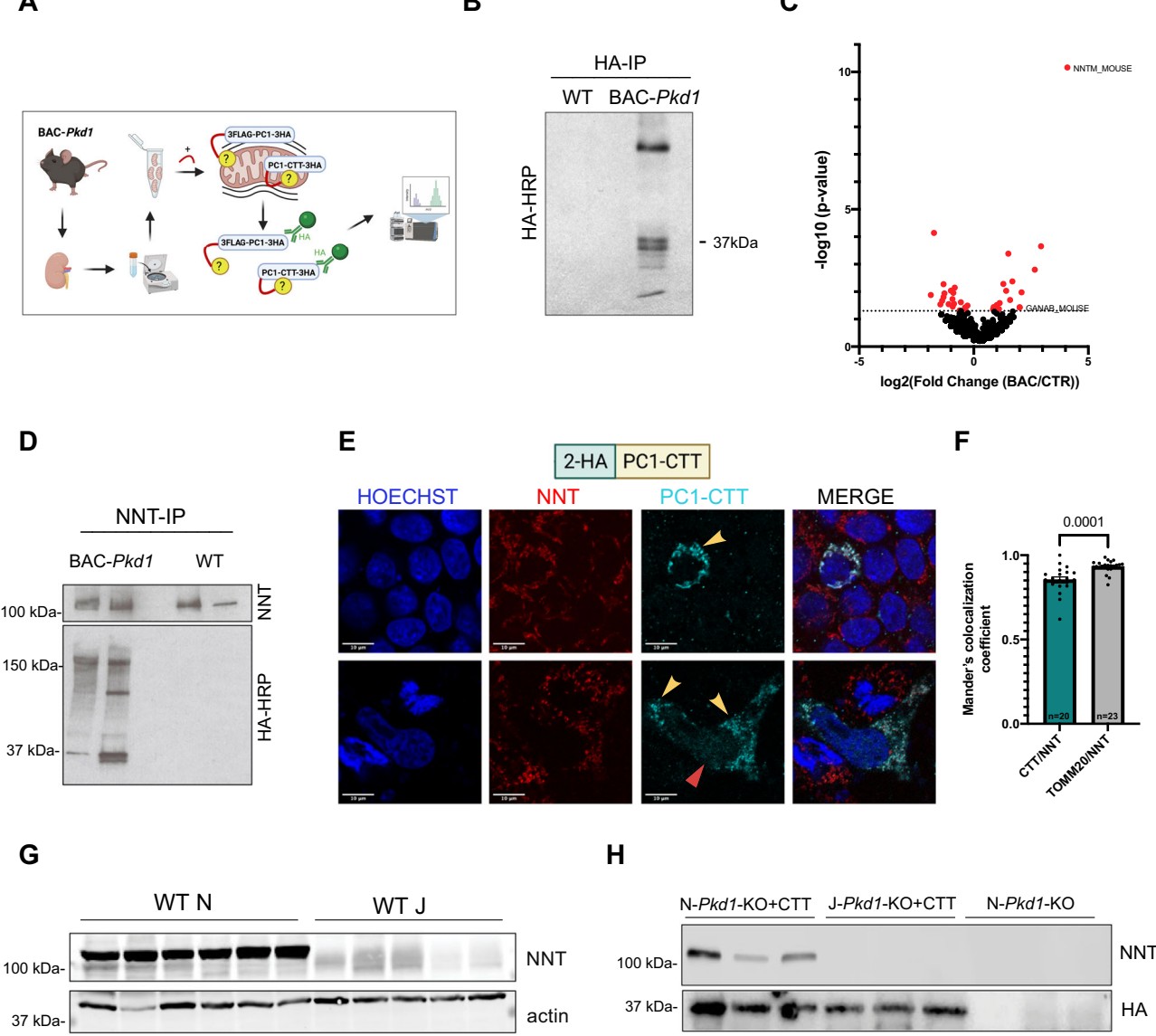

**Fig. 2 | 2HA-PC1-CTT (CTT) colocalizes and interacts with mitochondrial enzyme NNT. A** Workflow for identification of protein interactors of mitochondrial-associated pools of PC1 and its fragments. Mitochondria from *Pkd1*^F/^H^-BAC and WT mice were solubilized, crosslinked (3 mM DTSSP), and immuno-precipitated with anti-HA antibodies. Recovered proteins were identified by mass spectrometry. **B** Anti-HA immunoprecipitates from mitochondrial fractions were blotted with HRP-conjugated anti-HA, revealing recovery of the CTF of full-length 3FLAG-PC1-3HA as well as PC1-CTT-3HA fragments from *Pkd1*^F/H^-BAC but not WT control mice. **C** Volcano plot of PC1 and PC1-CTT interactors from *Pkd1*^F/H^-BAC vs WT kidneys (Colored dots: $P < 0.05$ determined by two-tailed Fisher's exact test; $n = 3$ per group). The amount of NNT co-immunoprecipitating with CTT from *Pkd1*^F/^H^-BAC kidneys was >16-fold greater than that in WT control immunoprecipitates ($P < 10^{-10}$). **D** Anti-NNT immunoprecipitates from kidney lysates were immuno-blotted with HRP-conjugated anti-HA, revealing coimmunoprecipitation of the 3FLAG-PC1-3HA CTF as well as PC1-CTT-3HA fragments from *Pkd1*^F/H^-BAC but not WT control mice. **E** Representative immunofluorescence (100×) image showing localization of the 2HA-PC1-CTT construct, identified with an anti-PC1-C-terminus

antibody, expressed by transient transfection in HEK293 cells. CTT colocalizes in mitochondria with endogenous NNT (yellow arrows). As previously reported[17], CTT was also observed in nuclei of a subset of transfected cells (red arrow). Scale bar:10 μm. **F** The mitochondria-associated fraction of CTT in (**E**) was assessed through Mander's colocalization analysis (20 individual cells from 14 independent images, 3 biological replicates), revealing a 0.8535 overlap coefficient, indicating extensive overlap between CTT and NNT distributions. As expected, the TOMM20/NNT colocalization coefficient (measured as an experimental positive control) is slightly but significantly greater. Data are shown as mean ± SEM. Pairwise comparison was performed using two-tailed Student's *t*-test. Colocalization of TOMM20 and NNT is shown in Supplementary Fig. 3. **G** Immunoblotting of total kidney lysates from WT "N" and "J" mice, confirming the presence and absence of NNT, respectively. **H** Immunoblotting of anti-HA immunoprecipitates from mouse kidney lysates, revealing immunoprecipitation of CTT in both "N" and "J" *Pkd1*-KO + CTT mice. NNT coimmunoprecipitation is detected exclusively in immunoprecipitates from N-*Pkd1*-KO + CTT mice. See Supplementary Data 1 for the complete comparative proteomic analysis depicted in (**C**). Source data are provided as a Source Data file.

recombination analysis was performed on 10-week rather than 16-week mice to minimize the potential that it could be affected by con-sequences of cystic disease progression. Furthermore, the *Nnt* muta-tion appears to be the major allelic difference between C57BL/6J and C57BL/6N strains[33] and is the only candidate genetic variation that has

been directly associated with the metabolic[29,34,35], cardiologic[36,37] and renal[38] differences observed between them. It is important to note, however, that the *rd8* retinal degeneration mutant of *Crb1* is detected exclusively in the "N" mice and results in a recessive ocular phenotype. To our knowledge, this is the only other phenotypically significant

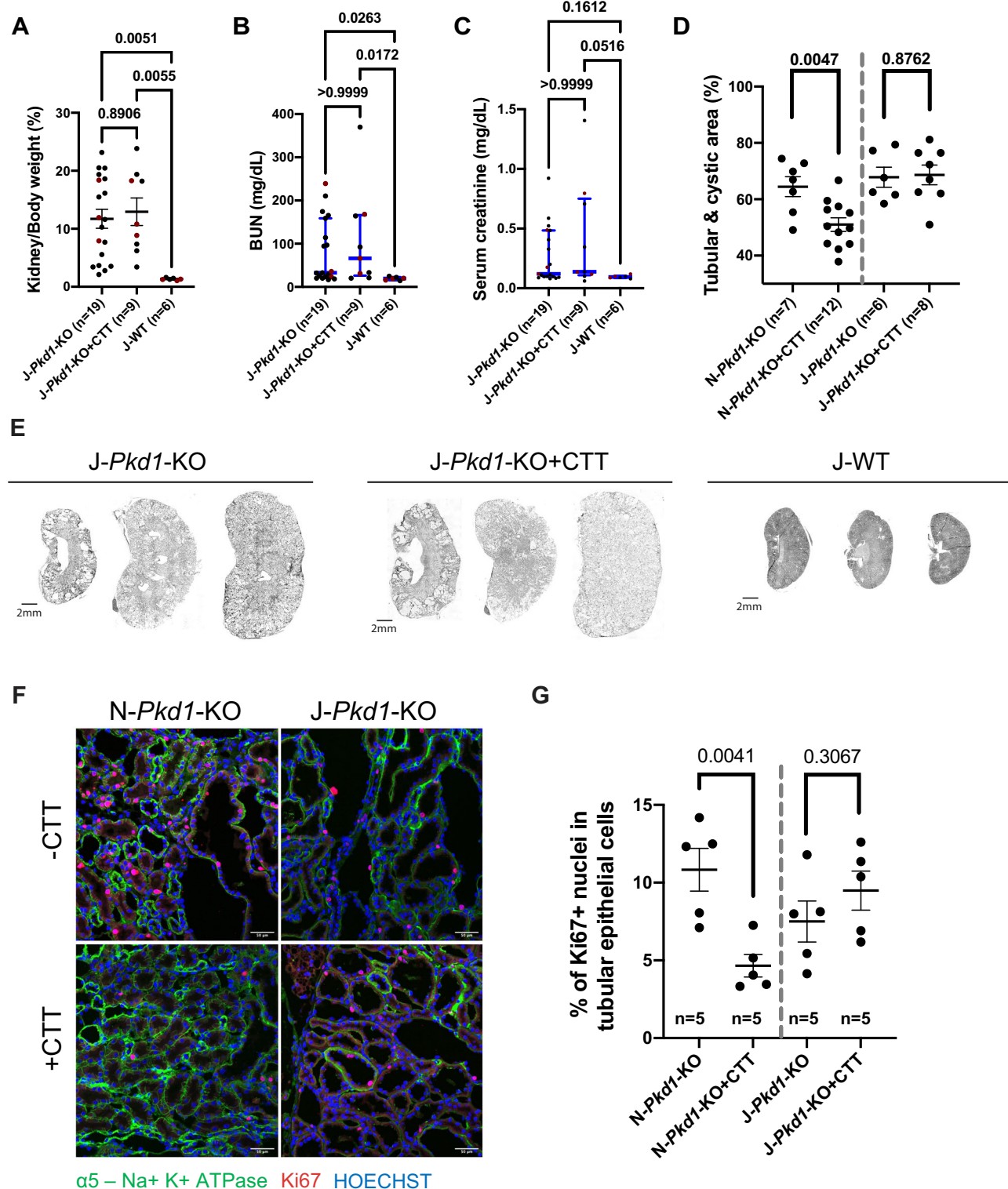

mutation that differs between "N" and "J" mice[33,39]. We found equal distribution of the *rd8* mutant allele in both N-*Pkd1*-KO + CTT and N-*Pkd1*-KO mice (Supplementary Fig. 6), thus excluding the possibility that skewed distributions of this mutant allele could account for the observed phenotypic difference between both groups. Additionally, this observation suggests that any other potential independently assorted N versus J allelic variants are unlikely to be responsible for the observed phenotype differences.

## Expression of PC1-CTT partially suppresses cystic disease in a rapidly progressive orthologous mouse model of ADPKD in an NNT-dependent manner

To assess the efficacy of NNT-dependent CTT suppression in an early and rapidly progressing model of cystic disease, we took advantage of mice in which Cre-expression is driven by the kidney collecting duct-specific *Pkhd1* promoter[40], which becomes active during embryogenesis. We crossed the previously characterized

**Fig. 3 | The interaction between NNT and CTT is critical to the CTT-mediated suppression of cystogenesis and tubular proliferation. A–C** Comparative analysis of *Pkd1*-KO + CTT and *Pkd1*-KO in the "J" background revealed no significant change in KW/BW ratio (**A**), BUN (**B**), and serum creatinine levels (**C**). Cystic mouse cohorts are composed of 53–55% female and 45–47% male mice. Red dots represent the animals depicted in (**E**). **D** Quantification of tubular and cystic area in H&E-stained kidney sections from *Pkd1*-KO and *Pkd1*-KO + CTT mice on both "N" and "J" backgrounds, as determined by ImageJ using images of renal cross-sections shown in Supplementary Fig. 1. **E** Representative H&E-stained kidney sections (4×) from 16-week J-*Pkd1*-KO, J-*Pkd1*-KO + CTT, and J-WT mice. Each group of sections is representative of the average KW/BW ratio of the entire cohort and is illustrative of the inherent variability associated with this mouse model[22,32]. Scale bar: 2 mm. **F** Representative immunofluorescence images (20×) showing tubular proliferation in *Pkd1*-KO and *Pkd1*-KO + CTT mice on "N" and "J" backgrounds, assessed by

Ki67 staining (red). Tubular epithelial cells are identified by positive Na,K-ATPase α-subunit staining (green). Scale bar: 50 μm. **G** Quantification of tubular proliferation determined by the percentage of Ki67-positive nuclei in renal tubular epithelial cells (*n* = 9 images per mouse, 5 mice per group). Counting of Ki67 positive nuclei was performed by an individual blinded to the experimental conditions. Parametric data are depicted with the mean ± SEM (black bars). Pairwise comparisons were performed using two-tailed Student's *t*-test (**D**, **G**). Multiple group comparisons were performed using one-way ANOVA followed by Tukey's multiple-comparisons test (**A**). Non-parametric data are depicted with the median and interquartile range (blue bars). Multiple group comparisons were performed using Kruskal-Wallis test followed by Dunn's multiple-comparisons test (**B**, **C**). H&E-stained kidney sections from all of the cystic mice included in this cohort are provided in Supplementary Fig. 1. Source data are provided as a Source Data file.

*Pkd1*<sup>fl/fl</sup>;*Pkhd1-Cre* (*Pkhd1-Cre;Pkd1*-KO)[22–24,41] ADPKD mouse model, generated on a C57BL/6J background, with N-*Pkd1*-KO + CTT mice (Fig. 4A). The resultant F1 progeny were heterozygous for WT *Nnt* (NJ). NJ;*Pkd1*<sup>fl/fl</sup>;*Pkhd1-Cre* (NJ;*Pkhd1-Cre;Pkd1*-KO) ± CTT littermates were evaluated at p14. We subsequently generated an F2 progeny by crossing F1xF1 NJ;*Pkd1*<sup>fl/+</sup>;*Pkhd1-Cre* mice. We then created two additional cohorts of cystic *Pkd1*<sup>fl/fl</sup>;*Pkhd1-Cre* mice by selecting these animals based on homozygosity for mutant or WT *Nnt* (JJ;*Pkhd1-Cre;Pkd1*-KO and NN;*Pkhd1-Cre;Pkd1*-KO, respectively). Littermates ± CTT were also evaluated at p14.

We found that CTT expression in F1 NJ mice partially suppressed cystic disease in this very early onset model of renal cystic disease, revealed through reduced KW/BW ratio at p14 (Fig. 4B, E and Supplementary Fig. 7a). More interestingly, we found that among F2 littermates, only those with the F2 NN genotype recapitulated the CTT-dependent phenotype suppression (Fig. 4C, F and Supplementary Fig. 7a). The F2 JJ littermates, generated from the same F1xF1 breeding as the F2 NN mice, did not show any CTT-dependent KW/BW change (Fig. 4D, G and Supplementary Fig. 7a). Levels of serum creatinine and BUN across all F1 NJ, F2 NN and F2 JJ remained unchanged (Supplementary Fig. 7b–d). This observation may reflect increased renal functional reserve in young mice[42]. In addition, the p14 endpoint is relatively early in the course of disease pathogenesis, considering a previously described 50th percentile survival rate of 31 days in this cystic mouse model[43]. While CTT expression produced no differences in tubular and cystic area relative to whole-kidney area in any of these 3 groups (Supplementary Fig. 7e), the total (absolute) tubular and cystic area in CTT-expressing NN and NJ mice was significantly reduced (Supplementary Fig. 7f), consistent with the decrease in tubular proliferation previously shown in CTT-expressing N-*Pkd1*-KO mice (Fig. 3F, G). No changes in this parameter were observed in CTT-expressing JJ mice (Supplementary Fig. 7f).

## PC1-CTT expression produces a marked NNT-dependent change in metabolic profile and modulates redox levels in 16-week-old N-*Pkd1*-KO mice with an established phenotype

In light of the localization of CTT to mitochondria and its NNT-dependent suppression of cystic disease, we evaluated potential metabolic consequences of CTT expression in *Pkd1*-KO mice by performing Liquid Chromatography Mass-Spectrometry (LC–MS) based metabolite profiling on whole-kidney tissue extracts across all 4 experimental groups in the adult model at 16 weeks. Principal Component Analysis (PCA) and hierarchical clustering revealed a distinct separation between *Pkd1*-KO and *Pkd1*-KO + CTT mice on the "N" background. In contrast, this analysis did not distinguish among these two groups on the "J" background (Fig. 5A). Forty-four metabolites met both criteria of *P* value <0.05 and fold change >2, 6 of them upregulated and 38 downregulated in N-*Pkd1*-KO + CTT compared to N-*Pkd1*-

KO kidneys (Fig. 5B), whereas comparison between J-*Pkd1*-KO + CTT and J-*Pkd1*-KO metabolic profiles revealed only very modest differences. Interestingly, many of the metabolites whose levels are reduced in N-*Pkd1*-KO + CTT mice have been previously implicated in ADPKD pathogenesis and some of them are related to potential therapeutic targets (Supplementary Fig. 8a, b) such as methionine[44], lactate[5,6], asparagine[45,46] and glutamate[45]. We also observed reductions in the levels of metabolites of the urea cycle, previously characterized as one of the most affected pathways in a pediatric ADPKD population[46]. The metabolic signature associated with CTT expression in 16-week-old N-*Pkd1*-KO mice is, therefore, marked by the reversal of dysregulated metabolites that are associated with ADPKD pathogenesis and progression. Of note, the cystic phenotypes of the animals used in the metabolomic studies were representative of those of the entire cohort shown in Fig. 1B–E and Fig. 3A–E (Supplementary Fig. 8c). Finally, in light of NNT's inherent capacity to modulate redox due to its role in catalyzing the exchange of reducing equivalents between NAD(P)(H) cofactors (Fig. 5C), we performed targeted LC–MS to quantify NAD(P)(H) levels in these same kidney homogenates. 16-week-old N-*Pkd1*-KO + CTT exhibited an increase in NADH/NAD⁺ and NADPH/NADP⁺ ratios when compared to N-*Pkd1*-KO mice (Fig. 5D, E), while CTT expression in J-*Pkd1*-KO mice did not affect either ratio.

## Mitochondria in N-*Pkd1*-KO + CTT mice exhibit increased NNT protein levels and increased assembly of ATP synthase and mitochondrial complex IV at a "per mitochondrion" level

We next determined whether CTT and CTT/NNT interactions alter the inventories of proteins that potentially affect mitochondrial function (Fig. 5F, G). Immunoblotting revealed a fourfold increase in NNT protein levels (NNT/actin) in 16-week-old CTT-expressing mice compared to N-*Pkd1*-KO littermates. This rise resulted from both increased mitochondrial mass (TOMM20/actin) and increased NNT protein levels when measured at a "per mitochondrion" level (NNT/TOMM20) (Fig. 5F, G). Comparable findings were observed in tissue from cystic human kidneys, which exhibited a significant decrease in NNT protein levels as compared to healthy kidney tissue (Fig. 5H). Based on their clinical course these ADPKD patients were presumed to manifest a *Pkd1*<sup>+/−</sup> genotype. Furthermore, western blotting employing a "mitococktail" antibody that interrogates the levels of stably assembled mitochondrial membrane complexes demonstrated increased levels of assembled ATP-synthase (complex V or CV) and cytochrome c oxidase (complex IV or CIV) at a "per mitochondrion" level, as revealed by an increase in both CV/TOMM20 and CIV/TOMM20 ratios in N-*Pkd1*-KO + CTT mice (Fig. 5F, G). No differences were observed in the assembly of complexes I (CI), II (CII) or III (CIII) in these same mice. The same comparison on the "J" background revealed no influence of CTT expression on mitochondrial mass or on mitochondrial complex assembly (Supplementary Fig. 9), suggesting that the effects observed in the "N" background involve the CTT/NNT interaction.

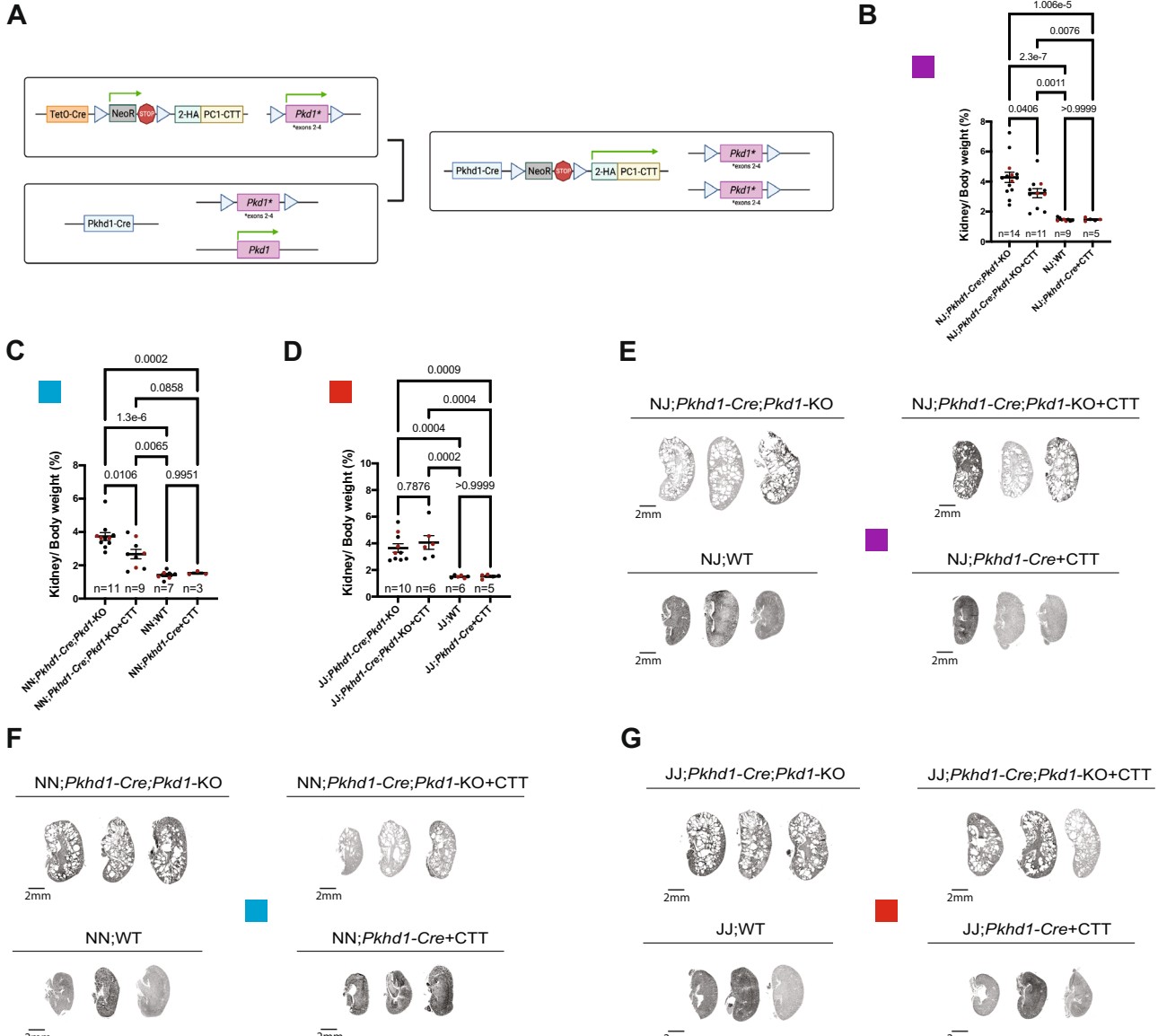

**Fig. 4 | Expression of CTT partially suppresses cystic disease in a rapidly progressive orthologous mouse model of ADPKD when generated on an NNT-competent background.** **A** Development of the *2HA-PC1-CTT;Pkd1^fl/fl^;Pkhd1-Cre* mouse model. We crossed the *2HA-PC1-CTT;Pkd1^fl/fl^;Pax8^rtTA^;TetO-Cre* mouse model on the "N" background *(N-Pkd1-KO + CTT)* with the previously characterized developmental *Pkd1^fl/fl^;Pkhd1-Cre* (*Pkhd1-Cre;Pkd1*-KO)[22–24,41] ADPKD mouse model, generated on the "J" background. Breeders with a *Pkd1^fl/+^;Pkhd1-Cre* genotype were used, due to the constitutive activity of the *Pkhd1-Cre* that initiates collecting duct-specific Cre-mediated recombination during embryonic stages and prevents animals with the complete *Pkd1^fl/fl^;Pkhd1-Cre* from reaching sexual maturity.
**B** Comparative analysis of NJ F1 (purple box) generation mice showing differences in KW/BW ratio. Of note, no phenotype differences were observed in NJ WT mice that do or do not express CTT (NJ;*Pkhd1-Cre* + CTT and NJ;WT, respectively). Red dots represent the animals depicted in (**E**). **C** Comparative analysis of NN F2 (blue box) generation mice showing differences in KW/BW ratio. Of note, no phenotype differences were observed in NN WT mice that do or do not express CTT (NN;*Pkhd1-Cre* + CTT and NN;WT, respectively). Red dots represent the animals depicted in (**F**). **D** Comparative analysis of JJ F2 (red box) generation mice showing differences in KW/BW ratio. Of note, no phenotype differences were observed in JJ WT mice that do or do not express CTT (JJ;*Pkhd1-Cre* + CTT and JJ;WT, respectively). Red dots represent the animals depicted in (**G**). **E**–**G** H&E-stained kidney sections (4×) from p14 NJ F1 (purple box), NN F2 (blue box), and JJ F2 (red box) mice. Each group of sections is representative of the average KW/BW ratio of the entire cohort. Scale bar: 2 mm. Data are depicted with the mean ± SEM. Multiple group comparisons were performed using one-way ANOVA followed by Tukey's multiple-comparisons test. H&E-stained kidney sections from all of the cystic mice included in these 3 cohorts are provided in Supplementary Fig. 7. Source data are provided as a Source Data file.

## PC1-CTT expression produces changes in NAD(P)(H) redox status and in metabolic profile at the 10-week pre-cystic time point

In light of the dramatic suppression of cystic disease severity that is produced by CTT expression in the 16-week N-*Pkd1*-KO mouse model, it is difficult to determine whether the distinct metabolomic profiles depicted in Fig. 5 arise as direct consequences of the CTT's potential influence on metabolism, or if they are instead secondary consequences of the very different degrees of cystic disease severity on the perturbations of metabolic processes that may be produced by disease progression. Hence, to further explore the direct effects of CTT expression on metabolism, we analyzed N-*Pkd1*-KO ± CTT mice prior to phenotype development at the pre-cystic 10-week time point[22] (Fig. 6A). As expected, no significant differences were observed in KW/BW ratio (Fig. 6B), serum creatinine (Fig. 6C) or BUN (Fig. 6D) in this 10-week cohort.

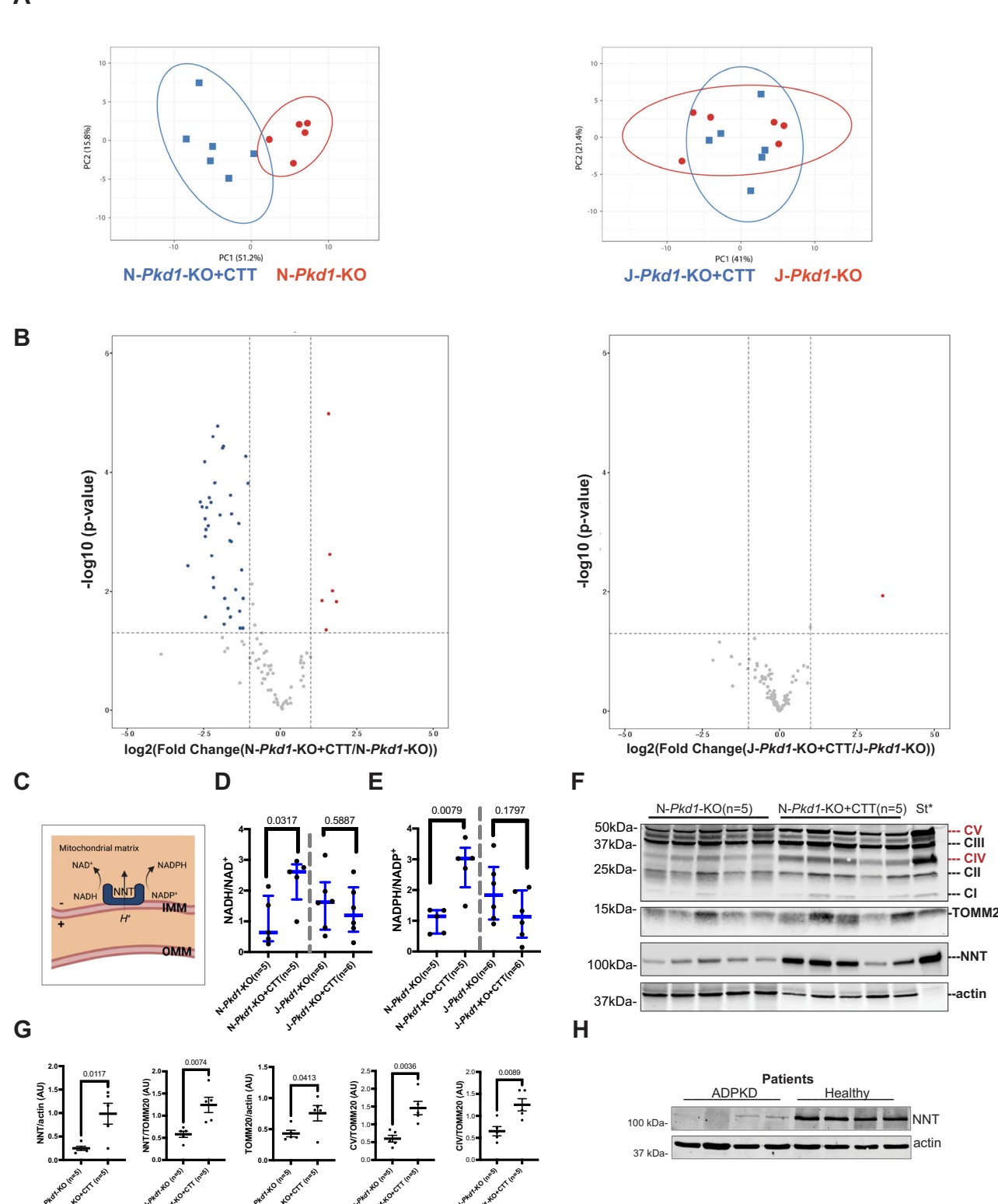

We performed LC–MS-based metabolite profiling on whole-kidney tissue extracts. Principal Component Analysis (PCA) and hierarchical clustering did not reveal a distinct separation between N-*Pkd1*-KO ± CTT (Fig. 6E); however, the levels of a dozen metabolites exhibited significant changes (*P* value <0.05 and fold change >2) at this time point, prior to the development of the pathological phenotype (Fig. 6F). These include purine nucleotide-related metabolites GDP,

ATP, and ADP, suggesting potential effects of CTT expression on the activity of purine and energy metabolism.

A striking feature of this untargeted metabolite profiling is that NADH and NADPH, both substrates of NNT, turned out to be among the most significantly increased metabolites upon CTT expression (Fig. 6F). A closer examination further revealed that levels of the oxidized cofactor NADP$^+$ are significantly reduced upon CTT expression

**Fig. 5 | CTT modulates redox and reduces ADPKD-associated metabolites in 16-week-old cystic mice. A** PCA plot of LC−MS metabolomic data from 16-week-old *Pkd1*-KO + CTT vs *Pkd1*-KO mice revealed group separation in the "N" but not "J" background. Marks report different samples (*n* = 5 or 6 mice/group); location in the plot is determined by relative contributions of metabolite subsets to variance. **B** Volcano plot showing metabolic profiling of kidney extracts from *Pkd1*-KO + CTT vs *Pkd1*-KO mice in the "N" and "J" backgrounds. Vertical lines mark twofold changes; horizontal lines mark *P* < 0.05 (two-tailed Student's *t*-test). Colored dots indicate metabolites with significant fold changes. Metabolite identification and labeling is provided in Supplementary Fig. 8a, b. *N* = 5 (N-*Pkd1*-KO) and *n* = 6 mice for N-*Pkd1*-KO + CTT, J-*Pkd1*-KO and J-*Pkd1*-KO + CTT groups. Animals used in these studies were representative of cohort phenotypes shown in Figs. 1B–D and 3A–C (Supplementary Fig. 8c). Complete untargeted comparative metabolomic analysis is provided in Supplementary Data 2. **C** Schematic depicting NNT in the inner mitochondrial membrane, using the mitochondrial proton gradient to catalyze hydride transfer between NADH and NADP$^+$ (forward enzymatic activity). **D**, **E** LC−MS detection of NAD(P)(H), showing significant differences in NADH/NAD$^+$

(**D**) and NADPH/NADP$^+$ (**E**) ratios between 16-week-old *Pkd1*-KO + CTT and *Pkd1*-KO mice exclusively in the "N" background. Data are normalized to mean values in N-*Pkd1*-KO mice and depicted with the median and interquartile range (blue bars). Pairwise comparisons were performed using two-tailed Mann-Whitney U test. **F**, **G** "Mitoccocktail" antibody immunoblots of N-*Pkd1*-KO + CTT and N-*Pkd1*-KO kidney lysates, reporting assembly status of mitochondrial complexes I, II, III, IV, and V. Blots were also probed with anti-TOMM20, anti-NNT and anti-actin (as loading control) (**F**). *St: rat heart mitochondrial extract, indicating positions of complexes I, II, III, IV, and V. Band densities were normalized to protein (actin) or mitochondrial content (TOMM20). Normalized band intensities within the same membrane are shown in graphs depicting the significant differences observed; *n* = 5/group (**G**). Additional mitochondrial mass and complex assembly assessments are in Supplementary Fig. 9. Data are depicted as mean ± SEM. Pairwise comparisons were performed using two-tailed Student's *t*-test. **H** Immunoblot depicting NNT levels in ADPKD and non-cystic human patient renal tissue. Source data are provided as a Source Data file.

with a fold change of 1.8 (less than the >2-fold cutoff). The changes in NADPH and NADH detected in the untargeted analysis prompted a targeted analysis for the NAD(P)(H) cofactors (Fig. 6G, H), which confirmed the results of the untargeted analysis, and revealed a sixfold increase in the NADPH/NADP$^+$ ratio and a fivefold increase in the NADH/NAD$^+$ ratio upon CTT expression. The changes in these redox ratios at the 10-week time point were even greater than those detected at 16 weeks, despite the absence of substantial metabolic and cystic phenotypes at this time point. These observations strongly suggest that the direct metabolic consequences of the CTT/NNT interaction include modulation of nicotinamide-related redox status, while many of the other metabolic phenotypes observed at the 16-week time point may be secondary consequences of this redox change or of cyst progression at later stages of the disease.

In light of the CTT/NNT localization to mitochondria, we hypothesized that the changes in redox levels observed in whole-kidney tissue would derive, at least in part, from changes in mitochondrial redox. To assess redox in the mitochondrial compartment, we generated an immortalized mouse tubular *Pkd1*$^{-/-}$ cell line that is derived from the collecting duct (Supplementary Fig. 10a). We transfected these cells with either the 2HA-PC1-CTT construct or the empty pcDNA3.1 vector (EV) control and isolated mitochondrial fractions from these live cells. CTT expression in this mitochondrial fraction was confirmed through immunoblotting (Supplementary Fig. 10b). We next performed a colorimetric NADH and NAD$^+$ assay that revealed increased mitochondrial NADH/NAD$^+$ ratios in CTT-transfected *Pkd1*$^{-/-}$ cells relative to those observed in EV-transfected *Pkd1*$^{-/-}$ cells (Supplementary Fig. 10c). Of note, EV-transfected cells exhibited a baseline mitochondrial NADH/NAD$^+$ ratio of ~0.3, within range of the mitochondrial NADH/NAD$^+$ ratios that are reported in literature (~0.1–0.2)[47], and significantly out of range when compared to the expected NADH/NAD$^+$ ratio in the cytosolic fraction (~0.001–0.01). These data indicate that our mitochondrial fraction isolation protocol was efficient and selective.

### PC1-CTT expression increases NNT activity ex vivo
While NAD(P)(H) measurements provide insights into the level of NNT function, these values cannot be interpreted as directly reporting NNT enzymatic activity since a large number of processes contribute to determining NAD(P)(H) levels. Thus, we evaluated NNT activity directly. To ensure that the assessment of NNT enzymatic activity was not influenced by the cystic phenotype, we conducted this experiment in a 10-week-old pre-cystic mouse cohort[22] (Fig. 7A, B).

Immunoblotting of a mitochondrial fraction from fresh kidney tissue revealed no significant differences in NNT expression among 10-week-old N-*Pkd1*-KO + CTT, N-*Pkd1*-KO and N-WT mice (Fig. 7C, D), in contrast to differences observed in 16-week-old animals (Fig. 5F, G).

The assessment of NNT enzymatic activity was performed using a standard kinetic spectrophotometric assay that detects NNT-mediated reduction of the NAD analog APAD[48]. We detected a 20% decrease in NNT enzymatic activity in N-*Pkd1*-KO mice compared to "N" WT controls (Fig. 7E, F). Furthermore, CTT expression in N-*Pkd1*-KO mice rescued NNT enzymatic activity to the same level observed in the healthy "N" controls (Fig. 7E, F). Of note, this assay was performed on mitochondria extracted from whole-kidney tissue, which includes multiple cell types in addition to Cre-expressing tubular cells. Hence, the magnitude of the observed difference is likely an underestimation of the true effect manifested in Cre-expressing cells.

### The capacity of the PC1-CTT to increase NNT enzymatic activity is dependent upon the presence of its putative mitochondrial-targeting sequence
We next wondered whether the previously identified and overlapping CTT nuclear localization[17] (aa 4134–4154) and mitochondrial-targeting[19] (aa 4129–4154) sequences (Fig. 7G) were required for the CTT-dependent increase in NNT enzymatic activity. To this end, we employed the previously described *Pkd1*$^{-/-}$ mouse tubular cell line (Supplementary Fig. 10a). We transfected these cells with 2HA-PC1-CTT (CTT), 2HA-PC1-CTTΔ4134−4154 (CTTΔ4134−4154) or the empty pcDNA3.1 vector (EV) (Fig. 7H) and employed the standard NNT kinetic spectrophotometric assay[48], optimized for this cell culture system. As expected, CTT transfection in vitro recapitulated our ex vivo findings obtained with mouse kidney tissue, significantly increasing NNT enzymatic activity by approximately 40% (Fig. 7I, J). In contrast, cells transfected with CTTΔ4134−4154 exhibited levels of NNT enzymatic activity that were similar to those observed in EV-transfected cells (Fig. 7I, J).

## Discussion
We report the unexpected finding that expressing the C-terminal 200 aa (CTT) of PC1 in an orthologous murine model of ADPKD is sufficient to suppress the development of the cystic phenotype. CTT expression resulted in dramatic preservation of renal function and morphology in an adult-onset mouse model of ADPKD, as evidenced by BUN and serum creatinine levels that are comparable to levels detected in healthy control mice. Slowing of cyst development as a consequence of CTT expression was also observed in a very rapidly progressing neonatal mouse model of the disease. Proteomic analysis performed on immunoprecipitates from a crude mitochondrial fraction revealed that NNT is the most significant PC1 binding partner in this setting. We confirmed that NNT interacts with CTT in vivo and showed that the CTT and NNT proteins colocalize in mitochondria when CTT is expressed in HEK293 cells. A previous study that employed a split GFP assay to assess PC1-CTT submitochondrial localization showed that

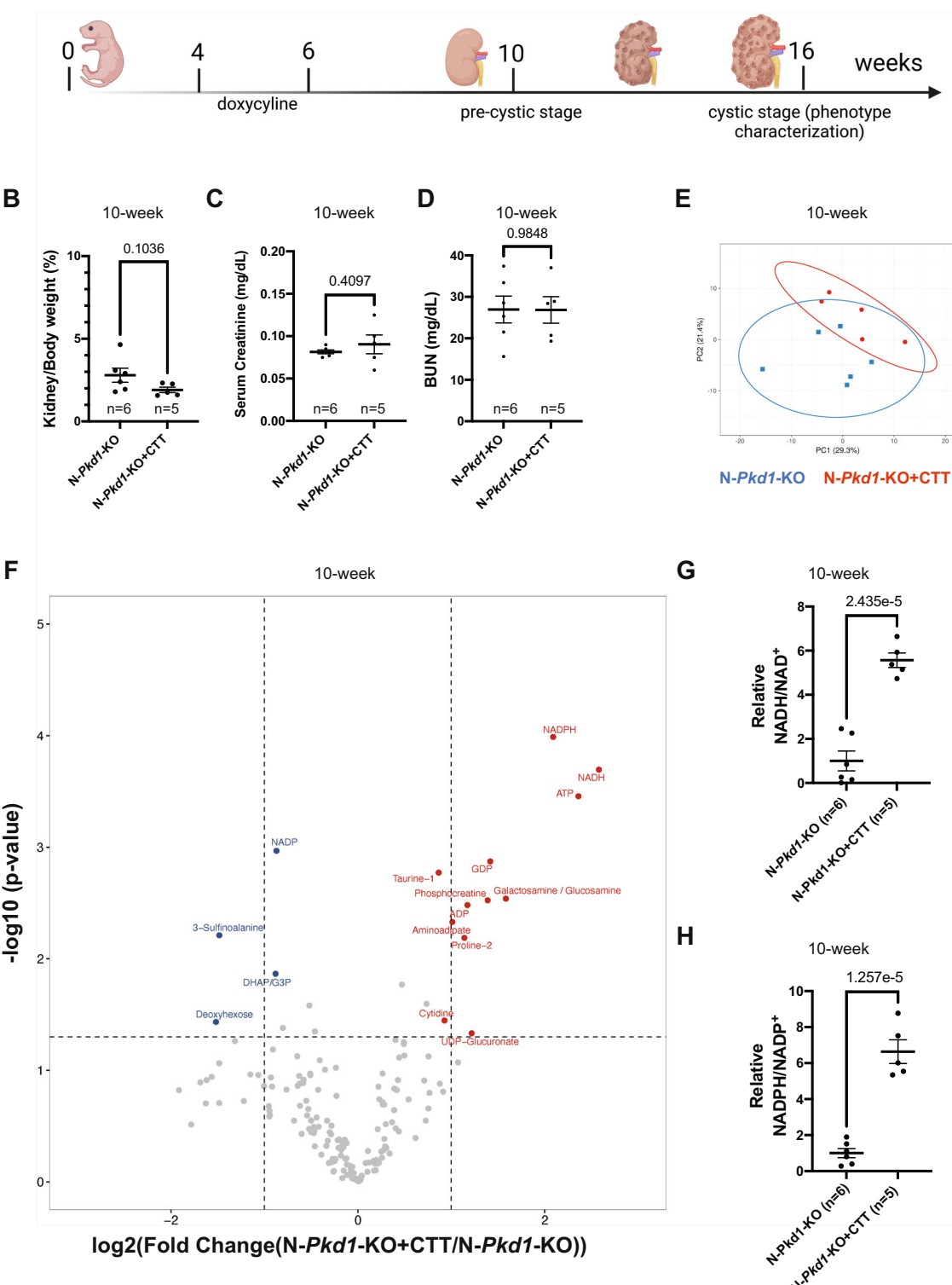

PC1-CTT localizes specifically to the mitochondrial matrix or matrix-facing surface of the mitochondrial inner membrane[19], a finding consistent with the predicted topological requirements of the interaction that we have identified between CTT and the mitochondrial matrix-facing inner membrane enzyme NNT. We showed that the suppression of the cystic phenotype produced by CTT expression in both the adult and neonatal models of ADPKD is dependent upon the availability of

this interaction, since no significant rescue is observed when CTT-expressing Pkd1-KO mice are compared to Pkd1-KO mice in the NNT-deficient "J" background. Similarly, tubular and cystic index and tubular proliferation parameters were only rescued in CTT-expressing Pkd1-KO mice generated in the "N" background. NNT enzymatic activity was reduced in the lysates from renal tissue of mice that lacked PC1. Expression of the CTT, both in vivo and in a cell culture system,

**Fig. 6 | CTT expression in pre-cystic mice reveals pronounced CTT-dependent redox modulation and distinct changes in metabolic profile. A** Schematic representation of the experimental timeline for assessment of untargeted comparative metabolomics in pre-cystic mice at 10 weeks of age. **B** Comparative analysis between pre-cystic N-*Pkd1*-KO ± CTT revealed no statistically significant change in KW/BW ratio. Cystic mouse cohorts are composed of 50–60% female and 40–50% male mice. **C, D** Kidney function is preserved in 10-week-old N-*Pkd1*-KO and N-*Pkd1*-KO + CTT mice, as revealed by normal and comparable values of serum creatinine (**C**) and BUN (**D**) in both groups. **E** PCA plot of LC–MS-based metabolomic data from 10-week-old N-*Pkd1*-KO + CTT vs N-*Pkd1*-KO. Each individual mark corresponds to a different sample (*n* = 5 or 6 mice per group as shown in figure) and its location in the plot is determined by the relative contributions of subsets of metabolites to the variance among samples. **F** Volcano plot showing differences in metabolic profiling of the kidney extracts from 10-week-old N-*Pkd1*-KO ± CTT mice. The vertical lines in each panel mark twofold changes; horizontal lines mark $P < 0.05$ determined by Student's *t*-test; *n* = 5 mice for N-*Pkd1*-KO + CTT and *n* = 6 mice for N-*Pkd1*-KO mice. The labeled colored dots indicate specific metabolites with significant fold changes. Complete untargeted comparative metabolomic analysis is provided in Supplementary Data 3. **G, H** LC–MS detection of NAD(P)(H) cofactors showing a ~5-fold change in NADH/NAD+ (**G**) and a ~6-fold change in NADPH/NADP+ (**H**) ratios in 10-week N-*Pkd1*-KO ± CTT mice. Data are normalized to mean values observed in N-*Pkd1*-KO mice. Data are depicted with the mean ± SEM. Pairwise comparisons were performed using two-tailed Student's *t*-test. Source data are provided as a Source Data file.

restored NNT enzymatic activity and this effect was dependent upon the presence within the sequence of the CTT of its putative mitochondrial localization sequence. Bulk kidney tissue metabolite profiling analysis performed on both cystic (16-week) and pre-cystic (10-week) N-*Pkd1*-KO + CTT and N-*Pkd1*-KO mice revealed changes in metabolic phenotypes, including NAD(P)(H) redox status at both time points. We showed that N-*Pkd1*-KO + CTT mice examined at the cystic 16-week stage exhibited significant and concomitant downregulation of multiple ADPKD-associated metabolites that have been identified over the past decade, and that have led to the suggestion of several metabolism-related potential therapeutic interventions for ADPKD. We assessed the effects of CTT expression on the stable assembly of electron transport chain (ETC) complexes. Previous studies have shown that increased quantities of stably assembled ETC complexes correlate with increased ETC activity and increased levels of oxidative phosphorylation[49,50]. We showed that CTT expression in N-*Pkd1*-KO mice not only leads to increased mitochondrial mass, but also to increased stable assembly of CIV and ATP-synthase after normalization to mitochondrial content. In concert with the metabolomic analysis, which revealed that expression of CTT on the "N" background leads to an increase in ATP levels in 10-week pre-cystic N-Pkd1-KO + CTT mice and to a fourfold reduction in lactate levels in these same mice at the 16-week time point (Supplementary Fig. 8a), these data are consistent with the interpretation that CTT expression induces a profound shift towards normal metabolism in which oxidative phosphorylation serves as the predominant source of ATP generation. Further experiments will be required to assess fully whether the observed metabolic changes do indeed reflect increased oxidative phosphorylation activity.

Taken together, our data suggest that expression of the PC1 C-terminal tail exerts effects that act upstream of the previously identified ADPKD-relevant metabolic pathways. Interestingly, in this context, a recent study suggests that the 200-aa PC1-CTT may play a crucial role in determining the prognosis for renal survival in human ADPKD patients[51]. This study examined 338 ADPKD patients from 82 pedigrees carrying non-truncating *PKD1* mutations. The mean renal survival in patients with *PKD1* mutations localized upstream of the GPS was 70.2 years, while the mean renal survival of non-truncating mutation carriers with mutations contained within the transmembrane domains or within the 200-aa CTT was 67.0 and 50.1 years[51], respectively. These data are consistent with the contention that CTT integrity may account for an approximately 20-year increase in renal survival in human patients and furthermore that the CTT may perform key physiological functions that contribute to suppressing cystic disease.

Interestingly, the presence or absence of NNT expression alone does not dramatically alter the disease course in *Pkd1*-KO mice. In the absence of CTT expression, both "N" and "J" *Pkd1*-KO mice develop severe cystic disease, and our data suggest that the NNT-competent strain exhibits even more aggressive disease progression. While the morphological and physiological consequences of cystic disease are severe in both 16-week-old "J" and "N" *Pkd1*-KO mice, we detect a roughly twofold increase in serum creatinine levels in "N" vs "J" *Pkd1*-KO mice (Supplementary Fig. 11). This is especially interesting since

declining GFR constitutes a fairly late consequence of cystic disease progression[52]. In line with our findings, a recent study characterized the disease progression in a mouse model homozygous for a *Pkd1* hypomorphic variant on three different strain backgrounds: BalbC/cJ (BC), 129S6/SvEvTac (129) and C57BL/6J[53]. While this study does not identify a specific modifier or mechanistic link, it presents robust evidence suggesting that disease progression in C57BL/6J was less severe than in the BC or 129 mice, which are expected to be NNT-expressing.

We hypothesize that the more aggressive phenotype in N-*Pkd1*-KO mice may be attributable to NNT's key role as an antioxidative enzyme, due to its capacity to exploit the protonmotive force across the mitochondrial inner membrane to drive the regeneration of NADPH from NADP+ utilizing NADH as the electron donor. By facilitating the detoxification of the reactive oxygen species (ROS) that have been shown to accumulate in hyperproliferative ADPKD tissue[54], NNT may allow cystic tissue to overcome oxidative stress and thus to proliferate further. In fact, the possible contributions of NNT to the severity of conditions involving hyperproliferation have been recognized in the context of cancer. NNT knockdown, which leads to increased apoptosis and decreased proliferation, is being explored as a therapeutic approach in models of various neoplasms, including gastric cancer[55], adrenocortical carcinoma[56] and hepatic adenocarcinoma[57].

The finding that NNT expression alone does not protect against cyst formation but that CTT expression suppresses cyst formation in an NNT-dependent manner suggests that NNT acts as a disease modifier rather than as a primary participant in cystogenic pathways. We observed changes in redox metabolites and a significant increase in NNT enzymatic activity in the "N" CTT-expressing model compared to N-Pkd1-KO mice. To make substrates accessible to NNT in the kinetic assay that we employed, mitochondrial membrane integrity is disrupted, which eliminates the NNT-driving protonmotive force. Thus, the possibility that CTT influences the directional kinetics of the enzymatic reaction remains to be explored. While the CTT-dependent increase in NNT activity in its forward-mode could lead to increased NADPH levels and enhanced antioxidative defense, it is tempting to hypothesize that CTT favors increased "reverse-mode" NNT activity, a finding that has been reported in pathological conditions[36,37] and in NADPH-rich environments[58]. "Reverse-mode" NNT activity leads to increased oxidative phosphorylation by increasing both NADH levels and the magnitude of the proton gradient. In addition, "reverse-mode" NNT activity would be expected to impair anti-oxidant defense[36]. This effect could suppress cystogenesis by increasing the susceptibility of oxidatively stressed cyst epithelial cells to apoptosis-inducing oxidative damage[59]. Our observation that CTT-dependent changes in cellular redox are associated with increases in both NADH and NADPH levels is somewhat surprising and is not entirely consistent with the predictions of either the forward or reverse-mode models. It is possible that CTT expression leads to unanticipated NNT-related effects on additional pathways such as the tricarboxylic acid cycle, purine biosynthesis, or the pentose phosphate shunt, that could lead to increases in either NADH or NADPH, which would in turn lead to NNT-dependent increases in the complementary redox species. It is also possible that

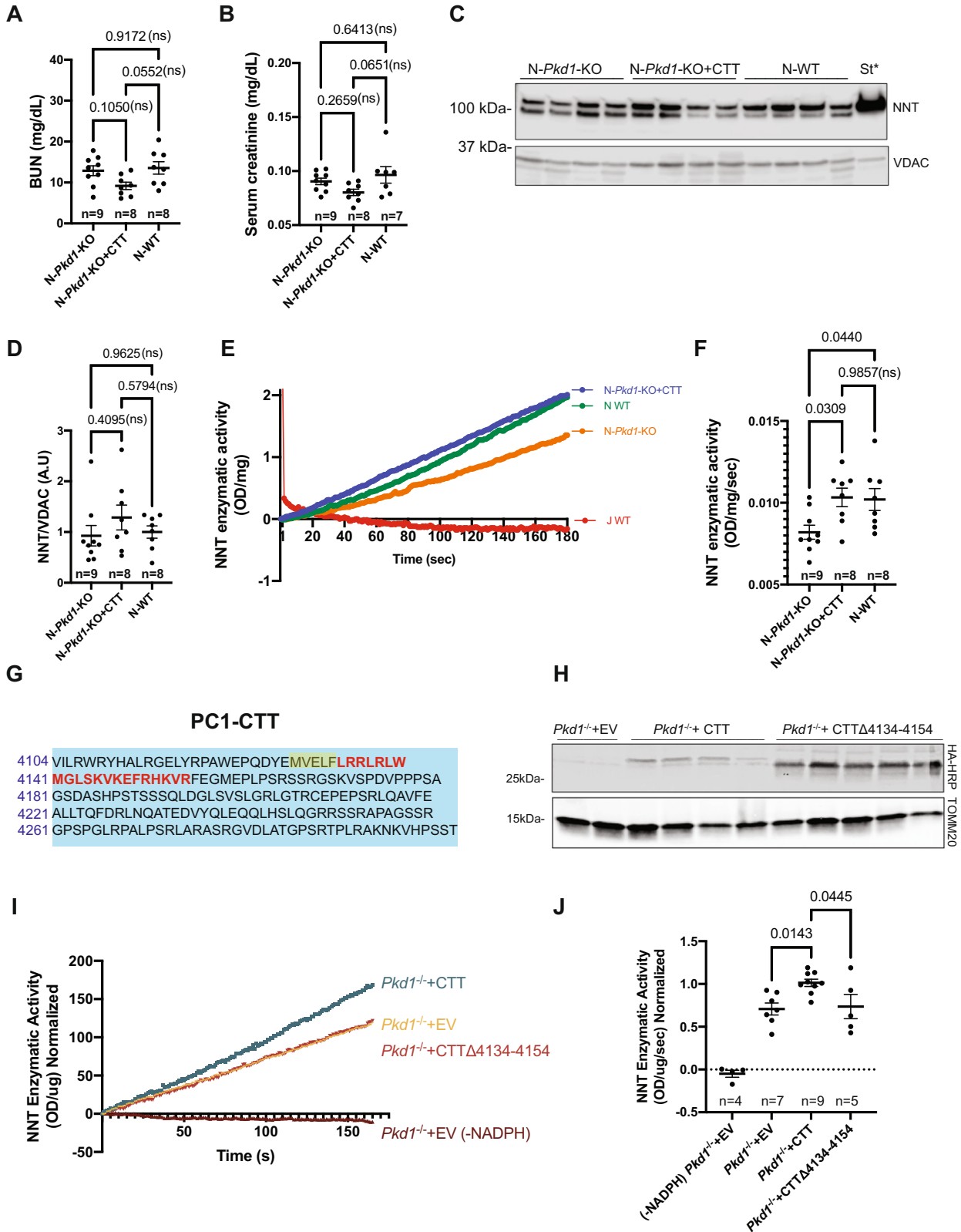

CTT expression, while increasing enzymatic activity, exerts no direct effect on the directionality of the enzymatic reaction, and that such directionality is determined solely by conditions of the renal microenvironment, such as NADPH availability and fluctuations in the mitochondrial proton gradient, that may vary in the course of the disease. In any case, further experiments will be necessary to fully define this mechanism.

In summary, we have identified an interaction between the mitochondrial enzyme NNT and the C-terminal tail of PC1 (Fig. 8). More importantly, we showed that expressing the 200 aa C-terminal tail of PC1 in a *Pkd1*-KO murine model of ADPKD is capable of significantly suppressing the cystic phenotype in an NNT-dependent fashion. Recent data shows that re-expression of full-length PC1 in a *Pkd1*-KO murine model results in rapid reversal of ADPKD[32]. While this finding

**Fig. 7 | CTT increases NNT activity in pre-cystic kidney tissue and cultured cells.**
**A**, **B** Kidney function is normal in all 10-week cohorts (N-*Pkd1*-KO, N-*Pkd1*-KO + CTT, and N-WT) used to assess NNT enzymatic activity, as revealed by BUN (**A**) and serum creatinine levels (**B**). Pre-cystic cohorts are composed of 45–50% female and 50–55% male mice. Data are depicted as means ± SEM. Multiple group comparisons were performed using one-way ANOVA followed by Tukey's multiple-comparisons test. **C**, **D** Immunoblot of mitochondrial extracts from N-*Pkd1*-KO, N-*Pkd1*-KO + CTT, and N-WT kidneys (**C**). *St: rat heart mitochondrial extract; VDAC served as mitochondrial loading control. NNT expression (normalized to VDAC) was not significantly different across all groups of 10-week-old mice (**D**). Data are depicted as mean ± SEM. Multiple group comparisons were performed using one-way ANOVA followed by Tukey's multiple-comparisons test. **E** NNT activity in mitochondria prepared from N-*Pkd1*-KO, N-*Pkd1*-KO + CTT, and N-WT kidneys, quantified by measuring rate of reduction of the NAD analog APAD. Samples were normalized to protein content. WT "J" mice served as negative controls, and confirmed the specificity of the assay by showing the expected absence of an upward slope.
**F** Comparison of NNT activity among N-*Pkd1*-KO, N-*Pkd1*-KO + CTT, and N-WT mice,

measured as ΔOD/s/mg of protein. Data are depicted with the mean ± SEM. Multiple group comparisons were performed using one-way ANOVA followed by Tukey's multiple-comparisons test. **G** Sequence of human PC1-CTT, highlighting the overlapping nuclear localization[17] (aa 4134–4154, red letters) and mitochondrial-targeting[19] sequences (aa 4129–4154, red letters and yellow highlight). **H** HRP-conjugated anti-HA immunoblotting of lysates from transfected *Pkd1*[-/-] cells revealing expression of both 2HA-PC1-CTT and 2HA-PC1-CTTΔ4134-4154. **I** NNT activity detected in mitochondrial extracts from *Pkd1*[-/-] cells transfected with 2HA-PC1-CTT (CTT), 2HA-PC1-CTTΔ4134–4154 (CTTΔ4134–4154) or empty pcDNA3.1 vector (EV). Absence of APAD reduction when NADPH is omitted from assay medium (-NADPH) confirms assay specificity. **J** Comparison of NNT activity among *Pkd1*[-/-] + EV, *Pkd1*[-/-] + CTT, and *Pkd1*[-/-] + CTTΔ4134–4154 mitochondrial extracts, measured as ΔOD/s/μg of protein and normalized to *Pkd1*[-/-] + CTT mean values. Data are depicted with the mean ± SEM. Multiple group comparisons were performed using one-way ANOVA followed by Dunnet's multiple-comparisons test, with *Pkd1*[-/-] + CTT serving as the reference group. Characterization of *Pkd1* cells is shown in Supplementary Fig. 10a. Source data are provided as a Source Data file.

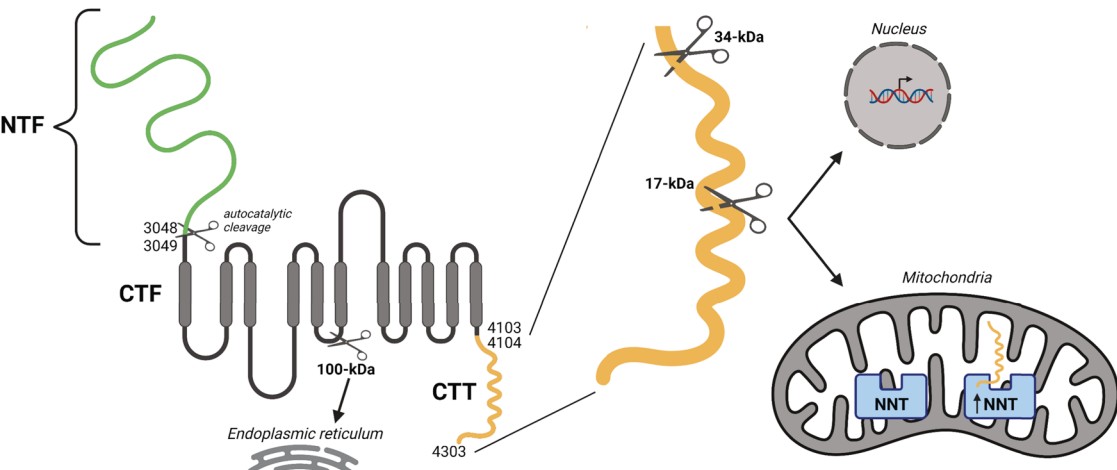

**Fig. 8 | Schematic representation and identification of PC1 cleavage sites and cleavage products.** The N-terminal domain of PC1 undergoes autocatalytic cleavage at the G protein-coupled receptor Proteolytic Site (GPS), generating a large 3048-aa N-terminal fragment (NTF) that remains non-covalently attached to the 1254-aa C-terminal fragment (CTF)[15]. PC1 undergoes further cleavage at the C-terminal domain, giving rise to PC1 CTF fragments, comprised of transmembrane domains and the cytoplasmic PC1 C-terminal tail. An ~100-kDa transmembrane CTF

localizes to the endoplasmic reticulum. Cleavage may occur within the C-terminal tail, generating PC1-CTT fragments ranging from 17 to 34-kDa that translocate to the nucleus and to mitochondria[17–19]. In the current study, we show that the 200 aa PC1-CTT can interact and increase the enzymatic activity of NNT in mitochondria, which produces a physiologically significant impact on mitochondrial redox, metabolism, and cystogenesis.

has broad theoretical and translational implications, the large size of the sequence that encodes the full-length PC1 protein severely limits its use in any conventional gene therapy approach. The observation that the PC1-CTT, which is only 200 residues in length, can suppress cyst formation suggests the very exciting possibility that its delivery via gene or protein therapy approaches could impact the course of the disease. It remains to be determined in future studies whether expression of the CTT fragment can recapitulate the ability of the full-length protein to reverse the course of established cystic disease. Whether or not this proves to be the case, the demonstration that the interaction between PC1-CTT and a component of the inner mitochondrial membrane suppresses cyst development opens new directions in the ongoing efforts to both understand the physiological functions of the polycystin proteins and to apply those insights to the development of new therapies.

## Methods
### Mouse models
All animal experiments were approved and conducted in accordance with Yale Animal Resources Center and Institutional Animal Care and

Use Committee (IACUC) regulations (protocol # 2019-20088). We utilized the previously characterized *Pkd1*[fl/fl];*Pax8*[rtTA];*TetO-Cre*[22], *Pkd1*[F/H]-BAC[23], and *Pkd1*[fl/fl];*Pkhd1-Cre*[22–24,41] mouse models. *Pkd1*[fl/fl];*Pax8*[rtTA]; *TetO-Cre* mice were generated on two distinct backgrounds by breeding in either C57BL/6J (stock no: 000664, Jackson Laboratories) or C57BL/6N (stock no:005304, Jackson Laboratories) strains. Additionally, we generated the *2HA-PC1-CTT; Pkd1*[fl/fl]; *Pax8*[rtTA];*TetO-Cre* on both C57BL/6J and C57BL/6N backgrounds. Animals were maintained at a 12:12 light:dark cycle, with 30–70% humidity and a 20–26 °C temperature. Animals were fed standard rodent diet (catalog #2018S, Teklad diets) ad libitum. In each experiment, animals were age-matched and sex distribution in sexually mature mice was similar across groups (specific descriptions in figure legends). Cre-negative littermates served as healthy WT controls.

### Generation of the *2HA-PC1-CTT;Pkd1*[fl/fl]*;Pax8*[rtTA]*;TetO-Cre* mouse mode
A *2HA-PKD1-CTT* BAC construct, which encodes a protein corresponding to a 2XHA tag linked to the N-terminus of the final 600 bp of human PC1, was generated with published BAC recombineering

technologies[32,60,61]. The cDNA sequence encoding *2HA-PKD1-CTT* was introduced into the *pRosa26-DEST* vector (catalog #21189, Addgene) such that a *lox-Neo(R)-3xSTOP-lox* is followed by the sequence encoding *2HA-PKD1-CTT*. A recombination cassette was constructed by flanking a *rpsL+-kana* selection cassette (catalog #20871, Addgene) with two same homology arms (1000 bp each arm) from *pRosa26-DEST*. Mouse Rosa26 BAC DNA was electroporated into DY380 bacteria that have stably integrated a defective $\lambda$ prophage containing the Red recombination genes *exo, bet* and *gam* under a strong *pL* promoter controlled by the temperature sensitive *cI857* repressor (kind gift from Dr. Donald Court, National Cancer Institute). The *rpsL+-kana* cassette was introduced into Rosa26 intron 1 region of the BAC after activation of the Red recombination system at 42 °C under positive selection by kanamycin resistance. The *rpsL+-kana* cassette in this intermediate was replaced by introducing *lox-Neo(R)-3xSTOP-lox* and *2xHA-PKD1-CTT* fragment with Rosa26 homology arms under negative selection with streptomycin sensitivity conferred by the *rpsL+* gene after the activation of the Red recombination system at 42 °C. The final *2xHA-PKD1-CTT Rosa26* BAC was shown to contain only the intended recombination and no other rearrangement using DNA restriction fingerprinting, direct sequencing and in vitro recombination in SW106 bacterial strain carrying an L-arabinose-inducible Cre gene.

Linearized modified BAC DNA purified by CHEF electrophoresis was used for pronuclear injection to generate transgenic founder lines. The BAC transgenic lines were produced in (C57BL/6J X SJL/J) $F_2$ zygotes. Founders were identified by PCR genotyping, verified by sequencing of PCR products and BAC copy number was determined by genomic quantitative PCR as described previously[23,24,32]. Two BAC founders with BAC copy numbers 2 or 4 were used in this study. All strains were backcrossed at least four generations with C57BL6 and are therefore expected to be at least 90% C57BL6 congenic. These animals were then crossed with *Pkd1^{fl/fl}; Pax8^{rtTA};TetO-Cre* mice to generate *2HA-PC1-CTT; Pkd1^{fl/fl};Pax8^{rtTA};TetO-Cre*, on both C57BL/6J and C57BL/6N backgrounds.

### Generation of the *2HA-PC1-CTT;Pkd1^{fl/fl};Pkhd1-Cre* mouse model

*2HA-PC1-CTT;Pkd1^{fl/fl};Pax8^{rtTA};TetO-Cre* mice, generated on the C57BL/6N background, were crossed with *Pkd1^{fl/+};Pkhd1-Cre* mice, generated on the C57BL/6J background. The F1 *Pkd1^{fl/fl};Pkhd1-Cre* progeny, heterozygous for WT *Nnt*, that did or did not express the 2HA-PC1-CTT BAC transgene, were analyzed comparatively at p14. We subsequently generated an F2 progeny by crossing F1xF1 mice. The F2 *Pkd1^{fl/fl};Pkhd1-Cre* mice were separated into 2 independent cohorts based on homozygosity for mutant or WT *Nnt*, and animals that did or did not express the 2HA-PC1-CTT BAC transgene were also analyzed comparatively at p14. Additionally, F1 and F2 *2-HA-PC1-CTT;Pkd1^{fl/+};Pkhd1-Cre* mice littermates were also generated and evaluated at p14 to assess whether any phenotypic differences were present in non-cystic mice that expressed the 2HA-PC1-CTT BAC transgene.

### Cell lines

**Mouse tubular Pkd1^{−/−} cell line.** Epithelial cells were collected from cystic renal tissue by standard methods[62]. Briefly, tissue obtained from an NN;*Pkd1^{fl/fl};Pkhd1-Cre* mouse was digested with 4 mg/ml of collagenase type I (catalog # SCR103, Sigma) for 1 h at 37 °C. The resultant renal cell suspension was seeded onto tissue-culture-treated plastic. After 4 days of growth, the mixed population of adhered kidney cells was transduced with a culture supernatant containing a retrovirus that drives expression of mTERT (catalog# 36413, Addgene). Retrovirus-containing culture supernatant was prepared according to the supplier's protocol. Following puromycin treatment to select for cells that had received the retrovirus, the surviving clonal colonies were propagated and kept in culture for 4 passages and then characterized through immunofluorescence labeling, western blotting and

quantitative PCR to assess the presence of markers of specific renal epithelial cell types.

**HEK293 cells.** HEK293 cells (cat# CRL-1573, ATCC) were cultured in DMEM supplemented with 10% fetal bovine serum (FBS), 1% penicillin/streptomycin, and 1% l-glutamine at 37 °C. These cells were then subjected to transient transfection following the protocol described in the transient transfection section of "Methods".

### Human specimens

The Baltimore Polycystic Kidney Disease Research and Clinical Core Center (P30DK090868) provided human kidney tissue from both ADPKD patients and non-affected controls. Samples were surgically harvested according to the guidelines established by the Institutional Review Board of the University of Maryland and were then de-identified. Sex information was not available. Immunoblotting with anti-NNT (catalog # 459170, Invitrogen) and anti-actin (catalog # A2228, Sigma) antibodies was performed using the protocol described in the Western blot section of Methods.

### Mouse kidney tissue harvest

Mice were euthanized according to Yale IACUC guidelines and following the recommendations of the American Veterinary Medical Association. Briefly, ketamine-xylazine solutions were prepared in 0.9% sodium chloride, containing a final concentration of 10 mg/ml of ketamine (NDC # 11695-0703-1, Covetrus) and 1 mg/ml of xylazine hydrochloride (catalog # X0059, TCI). All mice were euthanized with an intraperitoneal injection of 100 mg/kg of ketamine and 10 mg/kg of xylazine to produce deep anesthesia, followed by vital organ harvesting and exsanguination. Tail or toe tissue from previously genotyped mice was acquired for a second time when animals were under anesthesia for genotype confirmation. Retro-orbital blood was also collected from anesthetized mice, prior to kidney harvesting. The left kidney was excised, weighed, snap frozen in liquified $N_2$ and stored at −80 °C for biochemistry analysis. The right kidney was excised, weighed and fixed in 4% paraformaldehyde. Fixed kidneys were then sectioned in half along their sagittal axes, infiltrated with 30% sucrose overnight and embedded in OCT for further imaging.

### Serum creatinine and BUN measurement

Retro-orbital blood was collected from anesthetized mice prior to sacrifice and centrifuged in Plasma Separator Tubes with Lithium Heparin (BD) to separate plasma. Serum creatinine and BUN analysis were performed by the George M. O'Brien Kidney Center at Yale University.

### Immunoprecipitation

**Immunoprecipitation from crude mitochondria fractions prepared from *Pkd1^{F/H}-BAC mice.*** Kidneys from *Pkd1^{F/H}*-BAC and WT controls were isolated and homogenized with a Potter Elvehjem homogenizer, followed by a series of differential centrifugations at 4 °C according to the following protocol[25]: lysates were initially submitted to low-speed centrifugation at 740 × g for 5 min, and the collected supernatant was once again submitted to low-speed centrifugation at 740 × *g* for 5 min. The collected supernatant from this step was then submitted to high-speed centrifugation at 9000 × *g* for 10 min. The supernatant from this step was discarded and the pellet resuspended and submitted to high-speed centrifugation at 10,000 × *g* for 10 min. This final step was repeated twice, and the pellet containing crude mitochondria (both mitochondria and MAMs) was resuspended in PBS. These samples were incubated with 3 mM DTSSP (catalog# 803200-50MG, Sigma) to covalently crosslink interacting proteins at room temperature (RT) for 30 min in a rocking shaker and then the crosslinking reaction was quenched with 20 mM Tris-HCl pH 7.4. Samples were thereafter submitted to a high-speed 10,000 g centrifugation for 10 min and

resuspended in 100 μl of PBS + 1% SDS, followed by a 30-min immunoprecipitation with 25 μl of anti-HA magnetic beads (catalog # 88837, Thermo Fisher Scientific) and 4 washes with TENT buffer (10 mM Tris-HCl, 0.1 M NaCl, 1 mM EDTA, 5% v/v TritonX100). The final immunoprecipitate was eluted in 40 μl of 2× Laemmli sample buffer (catalog#1610747, Bio-Rad) with 300 mM DTT at 95 °C for 10 min for immunoblotting or proteomic analysis.

**Immunoprecipitation from kidney lysates prepared from *2HA-PC1-CTT;Pkd1^{fl/fl}; Pax8^{rtTA};TetO-Cre* mice.** Kidneys harvested from *2HA-PC1-CTT;Pkd1^{fl/fl};Pax8^{rtTA};TetO-Cre* mice in both "N" and "J" backgrounds and from *Pkd1^{fl/fl}; Pax8^{rtTA}; TetO-Cre* mice in the "N" background were snap frozen and stored at −80 °C. Homogenization was performed on ice using a motorized tissue grinder (catalog# 1214136, Fisher Scientific) in Tris lysis buffer (50 mM Tris pH 7.4, 100 mM NaCl, 0.5% NP-40, 0.5% Triton X100, 2 mM EDTA) supplemented with complete mini EDTA-free protease inhibitor cocktail tablets (catalog # 11836170001, Roche) and PhosSTOP phosphatase inhibitor cocktail tablets (catalog# 04906837001, Roche). Homogenates were then sonicated for 30 s (2 × 15 s bursts at 40% power) and incubated for 45 min on ice to complete protein solubilization. Lysates were centrifuged at 6000 × *g* for 15 min. Protein concentrations were measured with the Protein Assay Dye Reagent Concentrate (catalog #5000006, Bio-Rad). Anti-HA magnetic beads (catalog # 88837, Thermo Fisher Scientific) were equilibrated in lysis buffer (50 μl beads per reaction in 500 μl of lysis buffer) for 10 min at RT on a rocking shaker, and then incubated with a total of 500 μl of tissue lysate containing 4 mg of protein per sample, overnight, at 4 °C. Following four 5-min washes with 1 ml lysis buffer, the dynabeads were magnetically recovered and precipitated proteins were eluted in 60 μl of 2× Laemmli sample buffer (catalog#1610747, Bio-Rad) with 300 mM DTT. 1/3 of the eluted proteins (20 μl) was loaded per sample per gel for immunoblotting.

**Immunoprecipitation from kidney lysates prepared from *Pkd1^{F/H}-BAC* mice.** Kidneys from *Pkd1^{F/H}-BAC* and WT controls were snap frozen and stored at −80 °C. Immunoprecipitation from 500 μl of tissue lysate containing 4 mg of protein per sample was performed by adding 80 μl of agarose-conjugated anti-NNT (B3) (catalog # sc-390236 AC, Santa Cruz) according to the protocol described in "Immunoprecipitation from kidney lysates prepared from *2HA-PC1-CTT;Pkd1^{fl/fl}; Pax8^{rtTA};TetO-Cre* mice". A 1:100 ratio between primary antibody agarose conjugate and total lysate protein was maintained, following the manufacturer's protocol.

**Proteomic analysis**
Eluted proteins recovered by anti-HA antibody immunoprecipitation from the *Pkd1^{F/H}-BAC* kidneys were loaded onto an SDS-PAGE gel. In order to identify all interacting proteins, the SDS-PAGE gel was run only until the total sample volume had entered the gel before further separation happened, about 5 mm below the well. The gel was then fixed, and a band containing the entire sample was excised and subjected to mass spectrometric analysis performed by the Mass Spectrometry and Proteomics Resource of the W.M. Keck Foundation Biotechnology Resource Laboratory at Yale according to their standard operating procedures. The excised gel bands in 1.5 ml Eppendorf tubes were washed 4 times; first with 1 ml H₂O, then 1 ml 50:50 acetonitrile (CH₃CN):water, then 800 μl 50% H₂O/50% CH₃CN containing 100 mM NH₄HCO₃, then 800 μl 50% H₂O/50% acetonitrile containing 25 mM NH₄HCO₃. The final wash was removed, and the gel pieces were dried in a SpeedVac. Proteins in the gel were reduced with DTT at 37 °C for 30 min, then washed with CH₃CN and allowed to dry again. The proteins in the gel were then alkylated with iodoacetamide, then washed with 50% CH₃CN/50% H₂O containing 25 mM NH₄HCO₃, and were allowed to dry again. 100 μl of a 0.1 mg/ml stock solution of trypsin (Promega Trypsin Gold MS grade) with 20uL of 25 mM

NH₄HCO₃ was then added to the gel and digestion was carried out overnight, at 37 °C. The supernatant was then transferred to the injection vial for LC−MS/MS analyses.

Data dependent acquisition (DDA) LC−MS/MS data collection was performed on a Thermo Scientific Orbitrap Fusion connected to a Waters nanoACQUITY UPLC system equipped with a Waters Symmetry® C18 180 μm × 20 mm trap column and a 1.7-μm, 75 μm × 250 mm nanoACQUITY UPLC column (35 °C). To ensure a high level of identification and quantitation integrity, resolutions of 120,000 and 60,000 were utilized for MS and MS/MS data collection, respectively. MS and MS/MS (from Higher-energy C-Trap Dissociation (HCD)) spectra were acquired using a three-second cycle time with Dynamic Exclusion on. All MS (Profile) and MS/MS (centroid) peaks were detected in the Orbitrap. Trapping was carried out for 3 min at 5 μl/min in 99% buffer A (0.1% FA in water) and 1% buffer B [(0.075% FA in acetonitrile (ACN)] prior to eluting with linear gradients that reach 5% B at 5 min, 20% B at 90 min, 35% B at 125 min, and 97% B at 130 min for 5 min; then back down to 3% at 136 min. Four blank injections (1st 100% ACN, 2nd and 3rd 50:50 ACN:Water, and 4th buffer A) followed each sample injection to ensure there was no sample carryover. The LC−MS/MS Xcalibur ".raw" acquired data were processed with Proteome Discoverer (v. 2.0, Thermo Fisher Scientific) software and the protein identification was carried out using the Mascot search algorithm (Matrix Science, v. 2.6[63]). The data were searched against the SWISS-PROT Mus musculus protein database (2017), with peptide mass tolerance of ±10 ppm, fragment mass tolerance of ±0.02 Da, max missed cleavages of 2, Decoy setting-on, with the significant threshold at 0.05, and variable modification set, with oxidation of Methionine, carbamidomethyl of Cysteine, and propionamide of Cysteine. Proteins were considered identified when Mascot lists them as significant (false discovery rate of 1%) and more than 2 unique peptides match the same protein. The results were then uploaded to Yale Protein Expression Database (YPED[64]) for visualization, and further analyses were performed using the Scaffold Proteome Software 5.2.2.

**Western blotting**
Snap-frozen mouse kidneys were homogenized as described in the *2HA-PC1-CTT;Pkd1^{fl/fl};Pax8^{rtTA};TetO-Cre* whole-kidney lysate immunoprecipitation. Protein concentrations were measured with the Protein Assay Dye Reagent Concentrate (catalog #5000006, Bio-Rad). 20-40 μg of protein from whole-kidney lysate or 20 μl of IP eluted proteins were separated on 4−20% Mini-PROTEAN TGX Precast Protein Gels (catalog # 4561093, Bio-Rad) and electrophoretically transferred to a nitrocellulose membrane. Loading only surpassed the 20-40 μg range in the immunoblot depicted in Supplementary Fig. 2a, in which loading of 60 μg of whole-kidney lysate was necessary to identify PC1-CTT in both *2HA-PC1-CTT;Pkd1^{fl/fl};Pax8^{rtTA};TetO-Cre* and *Pkd1^{F/H}-BAC* mice. For western blotting of human renal tissue, homogenization was performed using a Polytron mechanical homogenizer in Tris lysis buffer (50 mM Tris pH 7.4, 100 mM NaCl, 0.5% NP-40, 0.5% Triton X100, 1 mM EDTA) for 15 s, at 300 rpm, on ice. Homogenates were then sonicated for 1 min, with 3 single continuous 15-s bursts at 40% power separated by a 5-s pause, left on ice for 60 min to complete protein solubilization, and centrifuged for 10 min at 10,000 × *g*. Membranes were sequentially incubated with blocking buffer (PBS, 6% (w/v) powdered milk/BSA, 0.1% Tween) followed by overnight incubation with primary antibodies. The primary antibodies used in this study were: anti-NNT (#459170, Invitrogen; #sc-390215, Santa Cruz), anti-NNT-HRP (#sc-390236HRP, Santa Cruz), anti-PC1-C-terminus (#EJH002, Kerafast), anti-HA-Peroxidase (#12013819001, Roche), Anti-HA-680 (#26183-D680, Thermo Fisher Scientific), anti-actin (#A2228, Sigma), anti-TOMM20 (#NBP1-81556, Novus Biologicals), anti-Total OXPHOS Cocktail (#MS604-300, Abcam) and anti-VDAC-HRP (#sc-390996HRP, Santa Cruz). All primary antibodies were used at a 1:1000 dilution, except for conjugated primaries anti-NNT-HRP (1:500), anti-HA-

Peroxidase (1:500), anti-HA-680 (1:500) and anti-VDAC-HRP (1:250). Unconjugated primary antibodies were detected using species-specific infrared (IR)-conjugated secondary IgG (1:5000; catalog #926-32211 and #926-68070, Li-Cor). Mitochondrial extract from rat heart tissue lysate (#ab110341, Abcam) was utilized as positive control. Membranes were visualized with either the Odyssey Infrared Imager (Li-Cor Biosciences) or Odyssey Fc (Li-Cor Biosciences) for chemiluminescence detection. Individual bands were quantified using ImageJ software (https://imagej.nih.gov/ij/, NIH).

### Transient transfection in cultured cells

We used Lipofectamine 2000 according to the manufacturer's protocol (catalog# 11668019, Thermo Fisher Scientific) to transiently transfect HEK293 and $Pkd1^{-/-}$ cells. Cells were transfected with the previously described 2HA-PC1-CTT and 2HA-PC1-CTTΔ4134-4154 constructs[21]. Briefly, the sequence encoding the final 200 aa of human PC1 (4104-4303) with an N-terminal 2xHA tag was cloned into the pcDNA3.1 zeo vector. The 2HA-PC1-CTT sequence is identical to that expressed in $Pkd1$-KO + CTT mice. Transfection with the empty pcDNA3.1 zeo plasmid was used as an experimental negative control.

### Immunofluorescence staining in cells

HEK293 cells grown on poly-L coated coverslips were fixed with 4% PFA in PBS for 30 min at RT followed by a 15-min treatment with permeabilization buffer (PBS, 1 mM MgCl$_2$, 0.1 mM CaCl$_2$, 0.1% BSA, 0.3% Triton X100). Cells were then blocked with goat serum dilution buffer (GSDB; 16% filtered goat serum, 0.3% Triton X100, 20 mM NaPi, pH 7.4, 150 mM NaCl) for 30 min, followed by a one-hour incubation with primary antibodies (1:100) diluted in GSDB. The primary antibodies utilized were anti-PC1-C-terminus (catalog #EJH002, Kerafast), anti-NNT (catalog #459170, Invitrogen) and anti-TOMM20 (catalog #NBP1-81556, Novus Biologicals). Following 3 PBS washes, samples were incubated with secondary antibodies (1:200) diluted in GSDB for one hour and then washed again with PBS. Alexa Fluor-conjugated antibodies (Alexa-594, 647; catalog #A11032 and #A31573 respectively, Life Technologies Invitrogen) were used as secondary reagents. Finally, coverslips were mounted on slides with VectaShield mounting medium (catalog# H-1000-10, Vector Laboratories) and imaged using a Zeiss LSM780 confocal microscope with the associated ZEN software version 3.6 blue edition. Images are the product of eightfold line averaging and contrast and brightness settings were chosen so that all pixels were in the linear range. This experiment was repeated three times. Mander's colocalization analysis was performed using Fiji 3-ImageJ (National Institutes of Health, Bethesda, MD) and Coloc 2-ImageJ plug-in. Briefly, the region of interest (ROI), defined as the non-nuclear area of a single transfected cell, was determined by tracing individual cells and subtracting all staining in Hoechst-positive areas. Staining for PC1-C-terminus and NNT were analyzed exclusively within the ROI and the overlap was assessed through Mander's colocalization analysis. Mander's colocalization analysis between TOMM20 and NNT, also in single cells, was used as a positive experimental control.

Immunofluorescence staining of cryostat sectioned cell aggregates of immortalized $Pkd1^{-/-}$ was performed using the immunofluorescence staining protocol described above. Primary antibodies used in the characterization of the $Pkd1^{-/-}$ cell line were anti-aquaporin-2 (C-17)(catalog# sc-9882, Santa Cruz), anti-NKCC2 (T9)[65], and anti-megalin (anti-MC-220)[66], all of which were used at a 1:100 dilution.

### Mouse tissue immunohistochemistry

Kidneys were fixed and processed for immunohistochemistry as described in the "mouse kidney tissue harvest" section of methods. Four-μm thick sections were heated in 10-mM citrate buffer for 15 min.

Slides were then blocked with 0.5% H$_2$O$_2$ in methanol for 30 min, followed by three 5-min 0.01 M PBS washes and further blocking with skim milk in PBS for 1 h at RT. Overnight incubation was performed with anti-NNT antibody (catalog# sc-390215, Santa Cruz) at a 1:50 dilution followed by detection using Vectastain Elite ABC-HRP kit (catalog# PK-6200, Vector Laboratories) according to the manufacturer's instructions.

### Proliferation assay

Kidneys were fixed and processed for immunofluorescence as described in the "mouse kidney tissue harvest" section of methods. Antigen retrieval for Ki67 was performed on 4-μm thick sections by heating slides in a 10-mM citrate buffer for 20 min. After a 30-min incubation with blocking buffer (PBS, 1%BSA, 10%goat serum), sections were co-incubated with anti-Ki67 (catalog# VP-RM04, Vector Laboratories) and anti-Na,K-ATPase α subunit (catalog# a5, DSHB) primary antibodies at a 1:100 dilution followed by detection with Alexa Fluor-conjugated secondary antibodies (Life Technologies Invitrogen) at a 1:200 dilution and Hoechst nuclear staining (catalog #H3570, Molecular Probes Invitrogen). Confocal images were obtained using a Zeiss LSM780 confocal microscope and the associated ZEN software version 3.6 blue edition. Images are the product of eightfold line averaging and contrast and brightness settings were chosen so that all pixels were in the linear range. Anti-Na,K-ATPase α-subunit was used as a tubular marker. Three images were acquired in the upper, middle and lower third of the kidney by a blinded investigator who also quantified the percentage of Ki67-positive nuclei relative to total tubular nuclei in these 9 independent images. A total of at least 2000 tubular nuclei were counted per animal.

### Morphological analyses

Whole-kidney images from hematoxylin and eosin-stained sagittal kidney sections were obtained at a ×4 magnification using automated image acquisition by the scan slide module in MetaMorph (Molecular Devices). The whole kidney was defined as the region of interest and the ImageJ default auto threshold function was employed to measure total cystic and tubular area, as well as cystic and tubular area relative to total kidney area by an individual blinded to experimental conditions.

### NNT enzymatic assay

Levels of NNT enzymatic activity were measured in N-$Pkd1$-KO, N-$Pkd1$-KO + CTT and N-WT mice according to a previously established protocol[48]. Briefly, 10-week-old pre-cystic[22] mice were euthanized according to Yale IACUC protocols. Left kidneys were extracted and partitioned in half in their coronal axis. One half of the kidney (approximately 70 mg) was used for mitochondrial preparation, which was performed using the Qproteome™ Mitochondria Isolation Kit (catolog# 37612, Qiagen). The Qiagen protocol was followed in detail until step 11a but the final wash (step 12a) was not performed in order to follow the suggested NNT-assay protocol[48]. Suspension of the final pellet was carried out with 20 μl of mitochondria storage buffer (catolog# 37612, Qiagen), a volume sufficient to allow the NNT assay and protein determination. Protein concentrations were measured with the Protein Assay Dye Reagent Concentrate (catalog #5000006, Bio-Rad), revealing a mean concentration of 7.64 ± 2.2 mg/ml. These values were within the protocols' 5–10 mg/ml predicted concentration. The assay medium was prepared and used within 24 h and was composed of 50 mM Tris-HCl (pH 8.0), 0.5% Brij 35, 1 mg/ml of lysolecithin and 300 μM of both NADPH and APAD. The assay was performed with 1 ml of assay buffer and 10 μl of the mitochondrial suspension and was read with a Bechman DU-640 UV-Vis spectrophotometer with a time-course setting: one measurement per s at a 375 nm wavelength, the chosen wavelength for reduced APAD. WT C57BL/6J and C57BL/6N were used as negative and positive controls, respectively. The former confirmed assay specificity (revealing no relevant activity) while the

latter confirmed sustained linear activity for several min, as reported in the original protocol[48]. This experiment was repeated three times. The protocol predicts occasional delays in activity initiation, reflecting the time taken for mitochondria to become permeable to substrates, and therefore an investigator blinded to genotype marked the starting point of the linear slope. Reactions were measured for a minimum of 150 s of optimal linear slope. The enzymatic activity was calculated by dividing the variation in optical density (OD; y axis) per variation in time (x axis, in s). Results are presented in activity per mg of protein, as established by the protocol[48].

This protocol[48] was further optimized for application to cell culture systems. Briefly, 100 µl microcuvettes were employed to reduce the total volume of both the mitochondrial suspension and assay buffer utilized for each reaction by a factor of 10. The protein concentration necessary to obtain optimal linear slopes was determined by serial titration of mitochondrial suspensions derived from $Pkd1^{-/-}$ cells, which revealed that 1 mg/ml was the lower threshold for confident detection of enzymatic activity. Experimental negative controls were obtained by omitting NADPH from the assay medium, thus preventing NNT-mediated reduction of APAD and confirming the specificity of the assay in this in vitro system. The assay was repeated with $Pkd1^{-/-}$ cells that had been transfected with 2HA-PC1-CTT (CTT), 2HA-PC1-CTTΔ4134–4154 (CTTΔ4134–4154) or the empty pcDNA3.1 vector (EV). Protein concentration of the mitochondrial extracts obtained from these cells ranged from 3.2–9.7 mg/ml. Reactions were assayed over an interval of a minimum of 150 s during which the slope was linear. Each reading was obtained from an independent transfection. The enzymatic activity was calculated by dividing the variation in optical density (OD; y axis) per variation in time (x axis, in s) and per protein concentration, and normalized to $Pkd1^{-/-}$ + CTT mean values.

### Genomic DNA isolation and quantitative RT-PCR

The DNeasy Blood & Tissue kit (catalog # 69504, Qiagen) was used to extract genomic DNA from all *2HA-PC1-CTT; Pkd1^fl/fl^;Pax8^rtTA^;TetO-Cre* and *Pkd1^fl/fl^;Pax8^rtTA^;TetO-Cre* mice included in the 16 and 10-week cohorts, starting with 20 mg of kidney tissue from each animal and following the manufacturer's instructions. Of note, we performed the optional 2-min treatment with 4 µl of RNAse A (100 mg/ml) at RT to obtain RNA-free genomic DNA in transcriptionally active tissues. Quantitative RT-PCR (qRT-PCR) was performed using iTaq Universal SYBR Green Supermix (catalog# 172-5121, Bio-rad). All samples were loaded in triplicates and reactions and data acquisition were performed using the Agilent real-time PCR system with its associated software. GAPDH levels were measured to normalize gene expression. Primers are listed below:

**Determination of *Pax8^rtTA^* and *TetO-Cre* copy numbers.** *Pax8^rtTA^* F: 5′-AAGTCATAAACGGCGCTCTG-3′

*Pax8^rtTA^* R: 5′-CAGTACAGGGTAGGCTGCTC-3′
*TetO-Cre* F: 5′-TCCATAGAAGACACTGGGACC-3′
*TetO-Cre* R: 5′-AGTAAAGTGTACAGGATCGGC-3′
*GAPDH* F: 5′-TGGTGTGACAGTGACTTGGG-3′
*GAPDH* R: 5′-GTCCTCAGTGTAGCCCAAGA-3′

Mouse samples were normalized to DNA obtained from a control mouse that expressed a single copy of both *Pax8^rtTA^* and *TetO-Cre*. All animals included in the present cohort presented a 1:1 or 2:1 ratio for both genes when compared to controls, confirming homozygosity or heterozygosity for both *Pax8^rtTA^* and *TetO-Cre* alleles.

**Determination of *Pkd1* rearrangement levels.** Both forward and reverse primers are situated within the floxed region (exons 2–4) of the *Pkd1* gene in the *Pkd1^fl/fl^;Pax8^rtTA^;TetO-Cre*. The primer sequences used were: *Pkd1*-F: 5′-TCTGTCATCTTGCCCTGTTCC-3′ and *Pkd1*-R: 5′-GTTGCACTCAAATGGGTTCCC-3′. The reverse primer is located in Chr17:24,783,583 (exon 4) and the forward primer is contained in the

prior intron at position Chr17:24,783,440. The amplified segment is therefore only present in the presence of intact WT *Pkd1*. *GAPDH* levels were measured to normalize gene expression. All pre-cystic mice were then normalized to the same 3 healthy controls and, as expected, exhibited lower expression of WT *Pkd1* compared to these WT controls. The ratio established between each cystic animal and WT controls served to define the rearrangement levels shown in Supplementary Fig. 5c.

### Metabolomics

LC/MS-based analyses were performed on a Q Exactive Plus benchtop orbitrap mass spectrometer equipped with an Ion Max source and a HESI II probe, which was coupled to a Vanquish UHPLC (Thermo Fisher Scientific). Polar metabolite extraction and detection methods were adapted from previous literature with minor modifications to be compatible with NAD(P)(H) measurement[67–70]. Specifically, 40 mg of snap-frozen tissue samples were ground using a mortar and pestle on dry ice. Metabolites were extracted with 800 µl 4/4/2 acetonitrile/methanol/water with 0.1 M formic acid, vortexed and incubated on dry ice for 3 min, and neutralized with 69.6 µl 15% ammonium bicarbonate. The samples were incubated on dry ice for 20 min, then centrifuged at $21,000 × g$ for 20 min at 4 °C, and 150 µl of supernatant were transferred to an LC–MS glass vial for analysis.

Polar metabolites were analyzed on Xbrige BEH Amide XP HILIC Column, 100 Å, 2.5 µm, 2.1 mm × 100 mm (catalog#186006091, Waters) for chromatographic separation. The column oven temperature was 27 °C, the injection volume 10 µl and the autosampler temperature 4 °C. Mobile phase A was 5% acetonitrile, 20 mM ammonium acetate/ammonium hydroxide, pH 9, and mobile phase B was 100% acetonitrile. LC gradient conditions at flow rate of 0.220 ml/min were as follows: 0 min: 85% B, 0.5 min: 85% B, 9 min: 35% B, 11 min: 2% B, 13.5 min: 85% B, 20 min: 85% B. The mass data were acquired in the polarity switching mode with full scan in a range of 70–1000 *m/z*, with the resolution at 70,000, with the AGC target at 1e⁶, the maximum injection time at 80 ms, the sheath gas flow at 50 units, the auxiliary gas flow at 10 units, the sweep gas flow at 2 units, the spray voltage at 2.5 kV, the capillary temperature at 310 °C, and the auxiliary gas heater temperature at 370 °C. Compound Discoverer (Thermo Fisher Scientific) was used to pick peaks and integrate intensity from raw data. The metabolite lists were filtered with minimal peak area > 1e⁷ (16-week samples) and 5e⁶ for >60% samples (10-week samples), and annotated by searching against an in-house chemical standard library with 5-ppm mass accuracy and 0.5 min retention time windows followed by manual curation. Finally, another 30 to 50 mg of tissue was obtained from each kidney used in the analysis and lysed in a 1:10 weight/volume ratio. We then performed a Bradford assay and normalized the data to tissue protein content. The PCA plots were generated by ClustVis (https://biit.cs.ut.ee/clustvis/) and the volcano plots using R script (https://www.r-project.org/).

NAD(P)(H) levels in the kidney extracts were analyzed on SeQuant ZIC-pHILIC polymeric 5 µm, 150×2.1 mm column (EMD-Millipore, 150460). Mobile phase A: 20 mM ammonium carbonate in water, pH 9.6 (adjusted with ammonium hydroxide), and mobile phase B: acetonitrile. The column was held at 27 °C, injection volume 5 µl, and an autosampler temperature of 4 °C. LC conditions at flow rate of 0.15 ml/min as follows: 0 min: 80% B, 0.5 min: 80% B, 20.5 min: 20% B, 21.3 min: 20%B, 21.5 min: 80% B till 29 min. The data were analyzed using the Xcalibur software.

### NADH and NAD$^+$ measurements in mitochondrial fractions

$Pkd1^{-/-}$ cells were transfected with 2HA-PC1-CTT (CTT) or the empty pcDNA3.1 vector (EV) and subjected to mitochondrial preparation using the Qproteome™ Mitochondria Isolation Kit (catolog# 37612, Qiagen). NADH and NAD$^+$ levels in these mitochondrial fractions were measured with the NAD/NADH Quantitation Kit (catalog# MAK037,

Sigma-Aldrich) following the manufacturer's protocol. Standard linear slopes from each individual experiment presented $R^2$ values above 0.96.

## Statistical analyses and reproducibility

Data quantification and plotting were performed using the GraphPad Prism software (https://www.graphpad.com/scientific-software/prism/), with the exception of metabolomic data, which were analyzed and plotted with ClustVis and R as described in the "Metabolomics" section of Methods, and proteomic data, which were analyzed and plotted with Scaffold and GraphPad Prism, respectively. Sample sizes for experiments involving *Pkd1fl/fl;Pax8rtTA;TetO-Cre* mice were chosen based on strategies used in previous analyses that have examined similar questions with the same experimental animal system[22,32]. Power calculations were performed prospectively prior to the generation of the F1 NJ;*Pkd1fl/fl;Pkhd1-Cre* (±CTT) cohort, based on the CTT-dependent phenotype suppression previously observed in the *Pkd1fl/fl;Pax8rtTA;TetO-Cre* model and on the observed variation in kidney-to-body weight ratios at p14 in cystic animals without CTT expression, which indicated that 12 animals per group would give 80% power to detect a 35% change in kidney-to-body weight ratio at a significance threshold of $P < 0.05$. Parametric data were depicted with the mean ± SEM. Non-parametric data were depicted with the median and interquartile range. Two-tailed Student's *t*-test or Mann-Whitney U test was used for pairwise comparisons, as indicated in the figure legends. One-way analysis of variance followed by Tukey's or Dunnett's multiple-comparison, or the Kruskal–Wallis test followed by Dunn's multiple-comparisons test was used for multiple comparisons, as indicated in the figure legends. $P < 0.05$ was considered statistically significant. The only mice excluded from the present study were those derived from the *Pkd1fl/fl;Pkhd1-Cre* cohorts sacrificed at p14 that were below the 3rd percentile of body weight. These animals were excluded in order to ensure that naturally occurring developmentally delayed runt pups[71] would not bias the present analyses. There is no correlation between the occurrence of runts and the *Pkd1fl/fl;Pkhd1-Cre* genotype[22–24,41]. Five out of the 101 total pups from the NJ F1, NN F2 and JJ F2 cohorts were excluded, one NJ;WT, one NJ;*Pkd1fl/fl;Pkhd1-Cre*, one NN;*Pkd1fl/+;Pkhd1-Cre* + CTT, one JJ;WT, and one JJ;*Pkd1fl/+;Pkhd1-Cre* + CTT. No other mice or data were excluded from the present analyses. All immunoblot data shown in Figs. 2, 5 and 7 are representative of three independent replicates.

## Reporting summary

Further information on research design is available in the Nature Portfolio Reporting Summary linked to this article.

## Data availability

The mass spectrometry proteomics data have been deposited to the ProteomeXchange Consortium via the PRIDE[72] partner repository under the dataset identifier: accession code PXD040210 (project webpage) and [ftp://ftp.pride.ebi.ac.uk/pride/data/archive/2023/02/PXD040210] (FTP download). The mass spectrometry metabolomics data have been deposited to the EMBL-EBI MetaboLights database[73] under the dataset identifier: accession code MTBLS7319. Source data are provided with this paper. The following databases/datasets were employed in the planning and analysis of experiments reported in this study: the SWISS-PROT Mus musculus protein database (2017), the mouse genome assembly GRCm39, the MitoCarta 2.0 mitochondrial proteome[26], the Human Protein Atlas[30] and the Rat Kidney Tubule Expression Atlas[31]. Source data are provided with this paper.

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

## Acknowledgements
This work was supported by the NIH grants DK120534 (M.J.C. and S.S.), DK072612 (M.J.C.), DoD grant PR191158 (M.J.C. and S.S.), and the PKD Foundation Research Grant 905682 (M.J.C.); by the Yale School of Medicine and Yale West Campus instrumentation support to H.S.; by the Yale School of Medicine startup funds and NIH grant R00 GM124296 to H.S.; and by a PKD Foundation Research Fellowship (215F20a) to L.O. We thank Dr. Hongyu Zhao (Prof. of Statistics and Data Science, Yale University) for generous guidance in designing the statistical analyses performed on all experiments. We thank SueAnn Mentone for outstanding technical assistance with immunofluorescence experiments on mouse tissue, Lonnette Diggs and the George M. O'Brien Kidney Center at Yale (NIH/NIDDK P30 DK079310) for BUN and serum creatinine measurements, and Jie Liu for assisting with tissue metabolite profiling. We also thank the Keck MS & Proteomics Resource at Yale School of Medicine for providing the necessary mass spectrometers and the accompanying biotechnology tools funded in part by the Yale School of Medicine and by the Office of The Director, National Institutes of Health (S10OD02365101A1, S10OD019967, and S10OD018034). The funders of this facility had no role in study design, data collection and analysis, decision to publish, or preparation of the manuscript. We thank Dr. Alessandra Boletta, Dr. Gerald Shulman, Dr. Leigh Goedeke, Brandon Hubbard and the entire Caplan laboratory for helpful discussions. We are extremely grateful to Drs. Luiz Fernando Onuchic, Michael Murphy, Ana C. Onuchic-Whitford, and Elieser H. Watanabe for their critical reviews and valuable contributions to the manuscript. Schematic Figs. 2a, 5c, 6a, 8 and Supplementary Fig. 5b were created with BioRender.com.

## Author contributions
L.O., V.P., K.D., X.S., H.S., S.S., and M.J.C. conceived and designed research. L.O., V.P., G.S., V.Rajendran, K.D., O.A., V.Rai, X.S., R.P., and N.P.G. performed experiments. L.O., V.P., G.S., R.P., X.S., H.S., S.S., and M.J.C. interpreted data. X.S performed bioinformatic analysis of metabolomic data. T.T.L and W.W gathered and performed the bioinformatic analysis of proteomic data. L.O., G.S., and V.Rai prepared the figures. L.O. and M.J.C. wrote the manuscript. All authors reviewed the manuscript and contributed valuable inputs.

## Competing interests
Some of the findings presented in this manuscript are included in the provisional patent application No 63/250,663 filed by Yale University that includes L.O., V.P. and M.J.C. as authors. The remaining authors declare no other competing interests.
