## [Peer Review File · Nature Communications]

The C-terminal tail of Polycystin-1 suppresses cystic disease in a mitochondrial enzyme-dependent fashionREVIEWER COMMENTS

Reviewer #1 (Remarks to the Author):

In their manuscript, Onuchic et al show that transgenic expression of a PC1 c-terminal fragment is able to suppress the cystic phenotype and preserve kidney function in genetic ADPKD mouse models and that this salutary effect depends on an interaction with the mitochondrial enzyme NNT. As the authors note, prior work had already shown that PC1-CTT translocate to mitochondria and that expression of CTT in Pkd1-KO cells rescues some of their altered mitochondrial function as well as their apoptotic/cystogenic phenotype. Therefore, the major advances in the current study are the demonstration that this mitochondrial trafficking impacts cystogenesis in vivo and the novel interaction with NNT. As the authors note, these findings raise the possibility that delivery of a ~200aa protein could be clinically beneficial. Overall, the approach is rigorous, using several elegant genetic approaches, including leveraging the 'natural' deletion of Nnt in C57BL/6J mice; the results are clearly conveyed; and the manuscript is well written. The mechanistic insight into how CTT's interaction with NNT impacts metabolism is somewhat underdeveloped, but the findings are nevertheless impactful and will be of interest to the journal's readership. I have several questions / comments:

Figure 2

The authors performed a proteomic analysis of anti-HA immunoprecipitates from Pkd1F/H-BAC mouse kidney mitochondrial fraction to identify NNT as a binding partner of PC1 and/or PC1-CTT. Subsequent work shows that PC1-CTT and NNT interact. But is an interaction between PC1 and NNT also possible? If a pull down is performed using anti-NNT antibody first, is only PC1-CTT detected?

Figure 5

The metabolomic analyses are difficult to interpret given the significant morphologic differences in kidneys by genotype by week 16, including the amount of kidney tissue. Based on the methods, 40mg of snap-frozen tissue were processed for these analyses. Does this translate into a comparable amount of total protein (i.e. per 40mg of tissue) in N-Pkd1-KO+CTT vs N-Pkd1-KO mice, for example as measured by Bradford or by assessment of B-actin etc? If the amount of living cellular content is lower per g in cystic kidneys, this would strongly bias towards lower levels of metabolites in these samples. Normalization to total protein may be more appropriate.

The authors note that CTT overexpression increases NNT 'expression', although I think they mean this in the broader sense of increasing NNT levels (whether it is by increasing expression, enhancing stability, reducing degradation, etc). Do the authors have any thoughts about how CTT and NNT interact?

Figure 6

NNT should have opposing effects on NADPH/NADP⁺ and NADH/NAD⁺ ratios, but CTT expression increases both. That said, these measurements were made on tissue extracts which combines the different cell types and different cellular pools of NADP⁺ and NAD⁺ (mitochondrial, cytosolic, nuclear). Understanding how CTT and NNT impact redox status in tubular epithelium, particularly in mitochondria and the cytosol is of interest and has the potential to provide insight on how they modulate oxidative metabolism and glycolysis. Can more specific, compartment-specific assessments of these measurements be made, for example in tubular Pkd-KO cells with or without CTT expression? These measurements could also provide insight on the directional kinetics of CTT-NNT.

Reviewer #2 (Remarks to the Author):

Laura Onuchic, Michael Caplan, and colleagues present data showing that re-expression of a c-terminal fragment of PC1 (PC1-CTT) can improve PKD in mice caused due to the lack of full-length PC1. This in-vivo work is based on earlier findings from the same group showing that PC1-CTT slows the proliferation of kidney cells in-vitro. They further report that PC1-CTT interacts with mitochondrial protein NNT, and suggest that this interaction facilitates NNT enzymatic activity. Finally, they show

that PC1-CTT re-expression fails to improve cystic disease in PKD1-KO mice in the 'J' background since these mice harbor a naturally occurring NNT deletion.

These results suggest an interesting idea that PC1-CTT can compensate for the loss of the full-length PC1. However, the rescue is not convincing in the Pax8 model and appears to be modest in the aggressive ADPKD model. If a fundamental function of PC1 CTT is to interact with NNT and modulate metabolic function, the rescue should occur irrespective of how aggressive the disease phenotype is. Another weakness is that the PC1-NNT link has not been conclusively validated. There are also conceptual gaps and inconsistencies in the analyses. Thus, the work – at this stage- appears preliminary.

Major comments:

1) The data from the PC1-CTT cross in the Pax8-Pkd1-KO model is not entirely convincing. First, the phenotypic variability in the N-Pkd1-KO model is immense. For example, the BUN range from the 20s (normal values) to the 300s (near-fatal kidney failure). Is ANOVA, which assumes a normal distribution, the proper statistical test for data with so much variability? How can any meaningful conclusions be drawn with so much variability?

2) The authors attempt to show equal Pkd1 recombination in N-Pkd1-KO and N-Pkd1-KO-CTT models. This analysis is very confusing. As I read it, relative expression of full-length PC1 is only marginally reduced in their cohorts compared to WT mice. In some cases, especially in the N-Pkd1-KO-CTT kidneys, there may be no recombination at all. If this interpretation is incorrect then the authors need to present their data more clearly. Additionally, they need to show how much downregulation of Pkd1 mRNA (exon-4) is observed in N-Pkd1-KO and N-Pkd1-KO-CTT models compared to non-recombined controls? Is there a correlation between the level of recombination and the observed phenotypic variability?

3) The PC1-CTT rescue in the Pkhd1/Cre;Pkd1F/F model is also not convincing. No analyses other than kidney to body weight ratios are provided. The authors should show cyst index, BUN, and creatinine. Did these improve as well? If a fundamental function of PC1 CTT is to interact with NNT and modulate metabolic function, the aggressiveness of the model should not be a huge issue. Why then is the rescue so modest?

4) The authors show that NNT enzymatic activity is reduced in the N-Pkd1-KO mice, which can be restored to WT levels by re-expressing PC1-CTT. This explains the PKD rescue by PC1-CTT. Using this same logic, since the "J" model lacks NNT and has zero enzymatic activity, you'd expect it to have a more severe phenotype than the "N" model. That does not appear to be the case. In fact, the J model seems to have lower BUN and creatinine than the N model. This argues against NNT playing a substantial role in cystogenesis.

5) The PC1-NNT physical interaction is superficially assessed. Does a reverse pull down with NNT immunoprecipitated PC1 CTT? What residues in PC1-CTT interact with NNT?

6) The functional PC1-NNT interaction is also not fully validated. Even though the authors attempt to point to NNT deletion as the sole difference between N and J models, there could be differences in SNPs, epifactors, etc. Thus, this interaction is not conclusively validated. The key experiment is to show that deleting NNT abolishes the PC1-CTT rescue in the "N" model?

7) The metabolomics (Figure 5) and ADPH/NADP+ and NADH/NAD+ experiments (Figure 6) were done at cystic timepoints. The authors should have assessed them at a precystic timepoint. The differences could simply arise due to the choice of samples used for analysis. It's also very important that the cystic phenotype (kidney weights, BUN, creatinine) of the samples used are shown. Were the samples representative of the cohort phenotype?

8) What is the relevance of reduced NNT expression in human samples? You may be comparing different cell types between ADPKD and healthy tissues. Did the authors find reduced NNT in mouse

models? IF data should be shown also.

Reviewer #3 (Remarks to the Author):

This is an interesting study addressing a very important translational question. Various lines of evidence suggest that mitochondrial function and metabolism in kidney tubules are altered in ADPKD, but mechanistic explanations were lacking. The authors show that overexpression of a fragment of PKD1 (CTT) offers significant protection in a mouse model of ADPKD. Moreover, they provide evidence that CTT interacts with a critical mitochondrial enzyme (NNT) and affects its function, and that the beneficial effect of CTT is lost in mouse lacking this enzyme ("J" type mice). These are intriguing and potentially important observations, which could open the way to new therapeutic strategies. The manuscript is clearly written and the experimental approach is elegant and impressive.

However, the major conclusion of the authors – that NNT mediates the protective effect - essentially rests on the lack of response in "J" type mice, and while this is certainly plausible, it is difficult to be absolutely certain that they are not resistant for another, as yet undiscovered reason. In this alternative scenario, changes in mitochondrial NNT function (and expression of other mitochondrial enzymes/complexes) in response to CTT might be consequential rather than causal. Moreover, another issue is that NNT expression seems to be predominantly in the distal tubular segments, but some of the metabolic changes observed in response to CTT appear to be more pertinent to the proximal tubule (PT). Resolving these issues would further strengthen this exciting study.

I have the following comments:

As mentioned already, the major mechanistic issue is whether the protective effects of CTT are specifically mediated via NNT. This could in principle be resolved through depletion of the enzyme in ADPKD mice. Alternatively, the authors performed in vitro experiments with cells overexpressing CTT. Could they demonstrate in the cell model that CTT directly alters mitochondrial function, and that this is dependent on NNT? The mechanistic importance of demonstrating this is underlined by the fact that the authors were not able to show mitochondrial localization of CTT in kidney tissue by antibody staining, due to technical issues.

The authors state in suppl fig 3 that NNT is mostly expressed in distal tubular segments. However, it is difficult from the images provided to clearly appreciate this. More zoomed in images with tubular segment and mitochondrial markers would be helpful. What is known about NNT expression along the mouse nephron from gene expression databases? Assuming that the expression is more distal, many of the impressive effects of CTT (e.g. suppression of cyst growth) seem to be more widespread. Moreover, when the authors perform metabolic studies on whole kidney tissue (e.g. metabolomic screening, western blotting for mitochondrial enzymes), the readouts are likely to be dominated by PTs. For example, the authors highlight changes in the urea cycle – to the best of my knowledge this is occurring in the PT. This raises the question as to whether the metabolic changes observed in CTT overexpressing mice are mediated via NNT, or are rather occurring downstream to some other protective effect(s). This point is further reinforced by the beneficial effect of CTT in a collecting duct specific model of ADPKD.

Following on the previous point, the authors could strengthen the notion that metabolic changes are happening specifically in the NNT expressing tubular segments by performing antibody staining to assess mitochondrial mass and complex expression.

Fig 3: Despite clearly developing cysts, the "J" type PKD1 KO mice do not appear to develop increases in BUN/creatinine compared to controls, as seen with the "N" type – do the authors have an explanation for this? Does it suggest that there may be baseline strain differences in response to PKD1 depletion?

Fig 3: The authors use Ki67 staining to assess cell proliferation, but it has been argued by others that this might not be an accurate readout in tubular cells (PMID: 29632300). Moreover, it would again be helpful here to identify tubular segments.

Fig 6: Why is the redox state of both NADH and NADPH shifted towards reduced? Wouldn't opposite changes in each be expected if NNT activity is increased? Did the authors observe a change in the total pool sizes of NADH and NADPH? In the preceding figure, the authors show a relatively low expression of CI in the kidney (as demonstrated previously by others, consistent with the notion that the majority of electron flux is probably via CII). If respiratory chain activity and electron flux through CII are increased in CTT overexpressing animals, might this cause an increase in reverse electron flow through CI, pushing the NADH pool towards a reduced state independent of NNT?

In their conclusions the authors state: "...these data support the interpretation that the CTT rescue model exhibits a profound shift towards oxidative phosphorylation as the predominant source of ATP generation". While the data are indeed consistent with this effect, the authors do not provide direct functional evidence to support the statement. As mentioned previously, this could be addressed with in vitro metabolic experiments (e.g. cells or isolated tubules). Or if not possible, the authors should at least clearly acknowledge this limitation.

Minor:

It could be helpful for readers to better understand the background to the study if the authors included a schematic of PKD1 structure and relevant fragments.

Fig 2: The authors should include TOM20 images to clearly demonstrate the mitochondrial location of CTT.

Abstract: The authors state that CTT overexpression "dramatically suppresses cystic phenotype". While the effect was clearly significant, it seems more of a partial effect? Please consider rewriting.

Reviewer #4 (Remarks to the Author):

Summary: Autosomal dominant polycystic kidney disease (ADPKD) is the most prevalent and potentially lethal monogenic disorder. Mutations in the PKD1 gene, encoding the PC1 protein, account for most cases. Post-translation proteolytic processing of PC1 generates a C-terminal fragment that is translocated to the mitochondria. The peptide comprising the 200 C-terminal amino acids suppresses the cystic phenotype in a PKD1 knockout mouse model of ADPKD. Proteomics data demonstrate this 200 aa fragment physically interacts with the mitochondrial enzyme nicotinamide nucleotide transhydrogenase (NNT). This interaction also impacts mitochondrial stoichiometry for TOMM20, complex V and cytochrome c oxidase. The C-terminal fragment inter Metabolomics data suggests that expression of this peptide in the PKD1 KO animals normalizes tissue levels of methionine, lactate, asparagine and glutamate among others. Data also suggest that the 200 C-terminal fragment interacting with NNT reduces the activity of NNT and diminishes the NADPH to NADP+ and the NADH to NAD+ ratio. Taken together the data suggest that the interaction of the 200 C-terminal fragment of PC1 interacts with NNT modifies the disease course and is not directly causally associated with the reduction in cystogenic pathways. It is encouraging that there is a potential for gene delivery of a smallish protein to mitigate the progressive course of a profound and prevalent disease.

Comments: Very elegant work to address the research question by developing competing animal models to subtract and add back in the PC1 protein and domains. Solid proteomics, metabolomics, immunofluorescence and immunoblot work.

(1) Previous pathways analysis of proteomics work demonstrated association of the 200aa C-terminal PC1 protein to the mitochondrial matrix. Crude mitochondrial preparations containing the mitochondrial matrix and mitochondrial-associated endoplasmic reticulum membranes was used for

identifying protein-protein interaction studies. These studies rely on DTSSP, a chemical crosslinker, to stabilize and enrich protein-protein interactions and specific binding partner data. The data demonstrate a specific enrichment in the interaction with NNT. Three missed opportunities exist here- (1) site specific identification of protein-protein interactions- DTSSP is a reducible chemical crosslinker. Proteolytic digestion of the DTSSP crosslinked sample without reduction & alkylation may present with an opportunity to identify crosslinked peptides. This method would require enrichment of digested peptides using a PD-10 type column and collection of data on +4/+5/+6 charge state ions and then analysis of the data by one of several available apps such as reviewed in Slavin M et al Anal. Chem. 2020, 92, 15899–15907. This might strengthen the information for the direct interactions of the C-terminus of PC1 with NNT or other mitochondrial matrix proteins. (2) why didn't the authors refine the volcano plot with PPI using Stringdb or other tools to understand how NNT might be interaction with other proteins in their samples. (3) What is the pathways analysis for these interactors.

(2) The authors use metabolomics based profiling of renal tissue to understand how the 200 C-terminal aa of PC1 impact renal parenchymal metabolism and identified dysregulated metabolomic features including three amino acids and lactic acid. While interesting I am not sure how this adds to the story. How does dysregulation of these amino acids tie in with the observed levels of reduced NADPH or NADH? Is this logical or a true-true but unrelated observation.

(3) Supplemental information on proteomics experiments is poor. Information on crosslinking mass spectrometry studies is completely absent. Given that the method for metabolomics is very detailed it would be essential to provide the same level of detail for the proteomics experiments. (Especially on how the crosslinking studies were conducted and MS data analyzed).

RESPONSES TO THE REVIEWERS' COMMENTS

We wish to sincerely thank all of the reviewers for their very thoughtful and insightful comments. We have taken the reviewers' critiques and suggestions very much to heart, and we have made extensive efforts to address their concerns. A large body of new data has been added both to primary and supplemental figures. We are happy to say that the addition of these data substantially addresses all of the points raised in the original review. We believe that the manuscript is much stronger and clearer thanks to the effort that was invested in order to incorporate the reviewers' suggestions and to correct the problems that they noted. We are grateful for the reviewers' help in further strengthening the novel findings that we present. We also wish to thank the reviewers for their patience. The experimental work that was needed to complete our revision proved to be more time-consuming than we had anticipated. We appreciate the willingness of the reviewers to consider the revised manuscript after our somewhat longer-than-intended delay in re-submitting it.

Reviewer #1 (Remarks to the Author):

In their manuscript, Onuchic et al show that transgenic expression of a PC1 c-terminal fragment is able to suppress the cystic phenotype and preserve kidney function in genetic ADPKD mouse models and that this salutary effect depends on an interaction with the mitochondrial enzyme NNT. As the authors note, prior work had already shown that PC1-CTT translocate to mitochondria and that expression of CTT in Pkd1-KO cells rescues some of their altered mitochondrial function as well as their apoptotic/cystogenic phenotype. Therefore, the major advances in the current study are the demonstration that this mitochondrial trafficking impacts cystogenesis in vivo and the novel interaction with NNT. As the authors note, these findings raise the possibility that delivery of a ~200aa protein could be clinically beneficial. Overall, the approach is rigorous, using several elegant genetic approaches, including leveraging the 'natural' deletion of Nnt in C57BL/6J mice; the results are clearly conveyed; and the manuscript is well written. The mechanistic insight into how CTT's interaction with NNT impacts metabolism is somewhat underdeveloped, but the findings are nevertheless impactful and will be of interest to the journal's readership. I have several questions / comments:

Figure 2

The authors performed a proteomic analysis of anti-HA immunoprecipitates from Pkd1F/H-BAC mouse kidney mitochondrial fraction to identify NNT as a binding partner of PC1 and/or PC1-CTT. Subsequent work shows that PC1-CTT and NNT interact. But is an interaction between PC1 and NNT also possible? If a pull down is performed using anti-NNT antibody first, is only PC1-CTT detected?

Thank you for your question and suggestion. We performed this immunoprecipitation utilizing agarose-conjugated anti-NNT (B3) (catalog # sc-390236 AC, Santa Cruz) on kidney lysates from Pkd1F/H-BAC and WT mice and show that both PC1-CTT and the ~150-kDa CTF of full-length PC1 coimmunoprecipitate with NNT (revised Fig. 2d). Of note, a second group has recently confirmed our findings by also identifying NNT as a PC1 binding partner (unpublished data, pre-print available at <https://www.biorxiv.org/content/10.1101/2022.04.08.487705v1.full.pdf>).

Figure 5

The metabolomic analyses are difficult to interpret given the significant morphologic differences in kidneys by genotype by week 16, including the amount of kidney tissue. Based on the methods, 40mg of snap-frozen tissue were processed for these analyses. Does this translate into a comparable amount of total protein (i.e. per 40mg of tissue) in N-Pkd1-KO+CTT vs N-Pkd1-KO mice, for example as measured by Bradford or by assessment of B-actin etc? If the amount of living cellular content is lower per g in cystic kidneys, this would strongly bias towards lower levels of metabolites in these samples. Normalization to total protein may be more appropriate.

Thank you for the question. We did indeed take the potential differences in protein concentration into consideration. Briefly, another 30 to 50mg of tissue was obtained from each kidney used in the analysis and lysed in a 1:10 weight/volume ratio. We then performed a Bradford assay and used these values to further normalize metabolomic data. This normalization had only been mentioned briefly in the Methods section and is now made clearer in this section. The absolute protein concentration of each sample has been provided below:

16-week mice	protein concentration (ug/ml)
N1	8901
N2	6333
N3	6948
N4	7997
N5	6961
N6	7277
N7	8537
N8	6904
N9	9592
N10	10071
N11	10880
N12	12137
N13	14321
N14	7140
N15	11538
N16	7531
N17	10343
N18	12807
N19	12412
N20	11882
N21	17407
N22	12961
N23	14928

Furthermore, the same normalization strategy was utilized for the new 10-week metabolomic experiment provided in this revised version of the manuscript. In this new

experiment it may be observed that variability in protein concentration is smaller since kidney morphology is similar at this earlier time point (revised Fig. 6).

10-week mice	[prot] ug/ml
PKD1	13085
PKD2	15879
PKD3	13815
PKD4	13398
PKD5	12716
PKD6	12934
P200_1	16399
P200_2	17280
P200_3	17280
P200_4	11917
P200_5	14266

Of note, the authors note that CTT overexpression increases NNT 'expression', although I think they mean this in the broader sense of increasing NNT levels (whether it is by increasing expression, enhancing stability, reducing degradation, etc). Do the authors have any thoughts about how CTT and NNT interact?

Thank you for this question. We have shown that NNT protein levels are indeed lower in tissue from cystic ADPKD patients (revised Fig. 5h) and in tissue from cystic 16-week *N-Pkd1*-KO mice (revised Fig. 5f,g). In mice at the 10-week timepoint, prior to the onset of cystogenesis, there was no significant difference in NNT protein levels (revised Fig. 7c,d). This makes it difficult to determine whether changes in NNT protein levels arise as direct consequences of the CTT and NNT interaction or if these changes are secondary to other pathways that are perturbed as the disease progresses. In this revised version, we show that CTT can increase NNT enzymatic activity (revised Fig. 7e,f,i,j) and that the capacity of the CTT to increase NNT enzymatic activity is dependent upon the presence of its putative mitochondrial-targeting sequence (revised Fig. 7i,j). We have further explored this interaction by performing the reverse NNT-immunoprecipitation (revised Fig. 2d) and evaluating CTT-specific changes in redox modulation and metabolism in pre-cystic mice, prior to the development of secondary phenotypes that could occur as a consequence of disease progression.

In summary, we have demonstrated that the integrity of the mitochondrial targeting sequence is required to establish the CTT-dependent functional increase in NNT enzymatic activity, and we are engaged in ongoing collaborative studies with structural biologists that are designed to identify protein interaction domains involved in the NNT/CTT interaction. We hope that the reviewer will agree with our strong feeling that these structural studies are beyond the scope of the present manuscript.

Figure 6

NNT should have opposing effects on NADPH/NADP⁺ and NADH/NAD⁺ ratios, but CTT expression increases both. That said, these measurements were made on tissue extracts which combines the different cell types and different cellular pools of NADP⁺ and NAD⁺ (mitochondrial, cytosolic, nuclear). Understanding how CTT and NNT impact redox status in

tubular epithelium, particularly in mitochondria and the cytosol is of interest and has the potential to provide insight on how they modulate oxidative metabolism and glycolysis. Can more specific, compartment-specific assessments of these measurements be made, for example in tubular Pkd-KO cells with or without CTT expression? These measurements could also provide insight on the directional kinetics of CTT-NNT.

Thank you for this question and suggestion. We have made efforts to explore this issue by generating an immortalized mouse tubular *Pkd1*^{-/-} cell line derived from an NN;*Pkhd1-Cre*;*Pkd1*-KO mouse kidney (revised Supplementary Fig. 10a). We then transfected these cells with the 2HA-PC1-CTT construct or the empty vector (EV; negative control) and isolated mitochondrial fractions from these cells, using the Qproteome™ Mitochondria Isolation Kit (catalog# 37612, Qiagen) (revised Supplementary Fig. 10b). We successfully measured relative NADH and NAD⁺ levels in these mitochondrial fractions by employing the NAD/NADH Quantitation Kit (catalog# MAK037, Sigma-Aldrich). The EV transfected cells revealed an average mitochondrial NADH/NAD⁺ ratio of ~0.3 with low variability between samples from independent transfections. This is within range of the mitochondrial NADH/NAD⁺ ratios described in the literature (~0.1-0.2)¹ and significantly out of range when compared to the expected NADH/NAD⁺ in the cytosolic fraction (~0.001-0.01), suggesting high selectivity in the mitochondrial fraction isolation. Furthermore, the mitochondrial fractions from 2HA-PC1-CTT-expressing cells revealed an average 6-fold increase in the NADH/NAD⁺ ratio when compared to the EV-transfected controls (revised Supplementary Fig. 10c). There was greater variability within the 2HA-PC1-CTT samples, likely related to changes in transfection efficiency, since each sample was obtained from an independent transfection. These findings suggest that the redox changes observed in bulk kidney tissue using targeted mass spectrometry are likely a product of changes in mitochondrial redox secondary to CTT expression, which further supports the importance of the NNT/CTT interaction.

It is important to note, however, that while NAD(P)(H) measurements provide indirect insights into the level of NNT function, these values cannot be interpreted as directly reporting NNT enzymatic activity since a large number of processes contribute to determining NAD(P)(H) levels. These experiments proved to be quite technically demanding, as they necessitated testing a number of commercial kits in order to identify which of them provided the most reliable readings, and generating enough cells to obtain sufficient mitochondria for measurements. Consequently, we focused our efforts on mitochondrial NADH and NAD⁺ measurements because we believed that they were more informative than mitochondrial NADP(H) levels, as it has been previously determined that isocitrate dehydrogenase and malic enzyme have a relative contribution of 70% and 8%, respectively, to NADPH regeneration, while NNT is responsible for only 22%². Future studies will explore the NADP(H) levels in the mitochondrial lysates from these cells and endeavor to assess the relative contributions of NNT to this parameter.

Reviewer #2 (Remarks to the Author):

Laura Onuchic, Michael Caplan, and colleagues present data showing that re-expression of a c-terminal fragment of PC1 (PC1-CTT) can improve PKD in mice caused due to the lack of full-length PC1. This in-vivo work is based on earlier findings from the same group showing

that PC1-CTT slows the proliferation of kidney cells in-vitro. They further report that PC1-CTT interacts with mitochondrial protein NNT, and suggest that this interaction facilitates NNT enzymatic activity. Finally, they show that PC1-CTT re-expression fails to improve cystic disease in PKD1-KO mice in the 'J' background since these mice harbor a naturally occurring NNT deletion.

These results suggest an interesting idea that PC1-CTT can compensate for the loss of the full-length PC1. However, the rescue is not convincing in the Pax8 model and appears to be modest in the aggressive ADPKD model. If a fundamental function of PC1 CTT is to interact with NNT and modulate metabolic function, the rescue should occur irrespective of how aggressive the disease phenotype is. Another weakness is that the PC1-NNT link has not been conclusively validated. There are also conceptual gaps and inconsistencies in the analyses. Thus, the work – at this stage- appears preliminary.

Major comments:

1) The data from the PC1-CTT cross in the Pax8-Pkd1-KO model is not entirely convincing. First, the phenotypic variability in the N-Pkd1-KO model is immense. For example, the BUN range from the 20s (normal values) to the 300s (near-fatal kidney failure). Is ANOVA, which assumes a normal distribution, the proper statistical test for data with so much variability? How can any meaningful conclusions be drawn with so much variability?

We thank the reviewer for these helpful questions and insightful comments. We performed Shapiro-Wilk normality tests on the KW/BW ratios from 16-week "N" and "J" *Pkd1*-KO, *Pkd1*-KO+CTT, and WT mice, which revealed that all these groups passed this normality test ($\alpha=0.05$) with the exception of the N-*Pkd1*-KO+CTT mice. Of note, the ROUT outlier test (Q=1%) identified 2 outliers in the N-*Pkd1*-KO+CTT, and the exclusion of these 2 animals would lead to a normal distribution in this group as well. In the interest, however, of maintaining full transparency and presenting the complete data set, we chose to not exclude any of these potential outliers from the study. We agree with the reviewer's point that, in light of the presence of a group with non-normal distribution, the parametric One-Way Anova is not the best statistical approach. Consequently, we have modified the statistical analysis in Fig. 1b and have employed the non-parametric Kruskal-Wallis test followed by Dunn's multiple comparisons test. We have also modified the statistical tests performed on serum creatinine & BUN levels in these Pax8 animals to the non-parametric Kruskal-Wallis test followed by Dunn's multiple comparisons test when non-normal distribution was observed. We thank the reviewer again for pointing this out to us, and are happy to say that statistical significance is maintained in light of these changes in statistical analysis.

Furthermore, the inherent variability of the model is acknowledged in all figure legends and taking this variability into consideration, 100% of the mouse kidneys included in these cohorts are shown in the Supplementary data. We do not believe that this variability limits our capacity to draw meaningful conclusions from our studies. In fact, the variability of the *Pkd1*-KO Pax8 model is well-established and major advances in the field have been predicated upon the use of this model³⁻⁷. As an example, in "Renal Plasticity Revealed through Reversal of Polycystic Kidney Disease in Mice" (Dong *et al*, Nature Genetics, 2021), the KW/BW ratio observed in 16-week Pax8 *Pkd1*-KO mice ranged from ~ 3-25%, similar to the values we observe. BUN levels also ranged from normal to near-fatal kidney failure values, and the authors acknowledge the variable severity of this fully penetrant model in the manuscript⁵. In "Cell-Autonomous Hedgehog Signaling is not Required for cyst

formation in ADPKD" (Ma *et al*, JASN, 2019), the BUN observed in 18-week Pax8 *Pkd1*-KO mice ranged from the same 20s to the near 300s mg/dL that we observe. Finally, this model recapitulates the extensive variability observed in human patients⁸ to the extent that even discordance in renal disease severity between relatives is a well-established feature of ADPKD⁹.

2) The authors attempt to show equal *Pkd1* recombination in N-*Pkd1*-KO and N-*Pkd1*-KO-CTT models. This analysis is very confusing. As I read it, relative expression of full-length PC1 is only marginally reduced in their cohorts compared to WT mice. In some cases, especially in the N-*Pkd1*-KO-CTT kidneys, there may be no recombination at all. If this interpretation is incorrect then the authors need to present their data more clearly. Additionally, they need to show how much downregulation of *Pkd1* mRNA (exon-4) is observed in N-*Pkd1*-KO and N-*Pkd1*-KO-CTT models compared to non-recombined controls? Is there a correlation between the level of recombination and the observed phenotypic variability?

Thank you for the question and comments. We undertook the evaluation of *Pkd1* recombination levels to document to the fullest extent possible that the models that we employed experienced equal levels of Cre-mediated *Pkd1*-KO. We recognize that performing this analysis in the late-stage 16-week mouse cohort may have been complicated by the extent of cystic disease development. Thus, we generated an entirely new sex-balanced pre-cystic 10-week cohort and assessed *Pkd1* rearrangement levels prior to phenotype development (revised Supplementary Fig. 5b,c). The relative expression of full-length PC1 in all 4 genotypes was similar and ~60% of that observed in WT mice, which is expected due to the contribution of non-tubular cells that are not targeted by Cre-mediated recombination and due to potentially incomplete doxycycline induction of the tubular cells. The upper threshold for experimental variability within WT mice is approximately 15% (revised Supplementary Fig. 5c). Of note, relying on genomic DNA qPCR is an accepted standard for Cre-mediated genomic rearrangement¹⁰.

3) The PC1-CTT rescue in the *Pkhd1*/Cre;*Pkd1*F/F model is also not convincing. No analyses other than kidney to body weight ratios are provided. The authors should show cyst index, BUN, and creatinine. Did these improve as well? If a fundamental function of PC1 CTT is to interact with NNT and modulate metabolic function, the aggressiveness of the model should not be a huge issue. Why then is the rescue so modest?

Thank you for the question and comments. We acknowledge that suppression in the *Pkhd1*-Cre;*Pkd1*-KO is not as impressive as in the *Pkd1*^{fl/fl};*Pax8*^{rtTA};*TetO*-Cre model. We have now, however, generated a new F2 generation in which the selected littermates were either homozygous for WT or mutant *Nnt* and show that we can recapitulate the partial phenotype suppression previously shown in the CTT-expressing *Nnt*-heterozygote model only in the F2 *Nnt*-homozygote CTT expressing mice (revised Fig. 4b-g). We believe different factors might be contributing to the less pronounced suppression phenotype in this model, including:

1- **P14 is an early time point to sacrifice and analyze these animals.**

In revised Supplementary Fig. 7, we measured, as suggested, BUN and serum creatinine levels in NJ, NN & JJ cohorts. We did not observe significant differences in these levels between any of the groups in any of these 3 cohorts (no significant difference was observed in *Pkhd1*-Cre;*Pkd1*-KO vs WT mice) potentially reflecting increased renal functional reserve in young mice¹¹, as well as the relatively early p14 endpoint. This endpoint consideration is especially relevant in light of a previously

described 50th percentile survival rate of 31 days in this cystic mouse model¹² (acknowledged in manuscript). Of note, while CTT expression produced no differences in tubular & cystic area relative to whole kidney area in any of these 3 groups (revised Supplementary Fig. 7e), the total (absolute) tubular & cystic area in CTT-expressing NN and NJ mice was significantly reduced, while no changes were observed in CTT-expressing JJ mice (revised Supplementary Fig. 7f). These data are consistent with the decreased proliferation in CTT-expressing mice shown in revised Fig 3f,g.

2- Initiation of cystogenesis time point

While Cre-mediated recombination and CTT-expression initiates on p28 in doxycycline-inducible *Pax8^{rtTA};TetO-Cre* mice, the same process starts during embryonic stages in *Pkhd1-Cre* mice. Previous studies have shown that inactivation of *Pkd1* in a tamoxifen-inducible ADPKD mouse model before p13 results in severely cystic kidneys within 3 weeks, whereas inactivation at p14 and later results in cysts only after 5 months¹³. In light of the well-established and substantial effects that the timing of *Pkd1* deletion exerts on the aggressiveness of different mouse models, it is quite possible that the mechanisms that drive the very rapidly progressive disease associated with early inactivation are not as susceptible to the suppressive effects of CTT expression and the CTT/NTT interaction.

4) The authors show that NNT enzymatic activity is reduced in the N-Pkd1-KO mice, which can be restored to WT levels by re-expressing PC1-CTT. This explains the PKD rescue by PC1-CTT. Using this same logic, since the “J” model lacks NNT and has zero enzymatic activity, you’d expect it to have a more severe phenotype than the “N” model. That does not appear to be the case. In fact, the J model seems to have lower BUN and creatinine than the N model. This argues against NNT playing a substantial role in cystogenesis.

Thank you for the question and comments. We wish to clarify that we do not suggest that NNT plays a direct or causative role in cystogenic pathways (included in manuscript). We believe, instead, that it acts as a disease modifier. Our data show that NNT is not itself a suppressor but, when influenced by CTT, it can exert suppressive effects. Although our work is focused primarily on CTT effects in ADPKD, we have added Supplementary Fig 11 in which we re-plot N-*Pkd1*-KO vs J-*Pkd1*-KO data and show that serum creatinine levels were increased in N-*Pkd1*-KO mice. We extensively explore the implications of this interesting finding in the “Discussion” section of the manuscript and believe that it sheds light on the mechanism underlying CTT-dependent phenotype suppression. In summary, we propose that NNT-dependent ROS detoxification could allow for further proliferation of cystic tissue, similar to that observed in several cancer models¹⁴⁻¹⁶. These findings are supported by a previous study that has shown that disease progression in C57BL/6J was less severe than in the BC or 129 mice, which are expected to be NNT-expressing¹⁷. This finding also suggests that NNT function needs to be increased by CTT and perhaps its functional properties modified even further (i.e., potential reversal of the catalyzed enzymatic reaction in this pathological condition) as we now discuss in depth in the “Discussion” section of this manuscript.

5) The PC1-NNT physical interaction is superficially assessed. Does a reverse pull down with NNT immunoprecipitated PC1 CTT? What residues in PC1-CTT interact with NNT?

Thank you for your question and this helpful suggestion. We performed this immunoprecipitation utilizing agarose-conjugated anti-NNT (B3) (catalog # sc-390236 AC, Santa Cruz) on kidney lysates from Pkd1F/H-BAC and WT mice and show that both PC1-CTT and the ~150-kDa CTF of full-length PC1 coimmunoprecipitate with NNT (revised Fig. 2d). Of note, a second group has recently confirmed our findings by also identifying NNT as a PC1 binding partner (unpublished data, pre-print available at <https://www.biorxiv.org/content/10.1101/2022.04.08.487705v1.full.pdf>). While the protein domains that contribute to the CTT/NNT interaction remain to be discovered, we performed new experiments that show that the influence of the PC1 CTT on the enzymatic activity of NNT is prevented by deletion of the 20 residues of the CTT that contain the putative mitochondrial localization motif (Fig. 8e,f). We share the reviewer's interest in learning more about the structural details of the interaction between the CTT and NNT. We are currently engaged in on-going collaborative studies with structural biologists that are designed to identify protein interaction domains involved in the NNT/CTT interaction. We hope that the reviewer will agree with our strong feeling that these structural studies are beyond the scope of the present manuscript.

6) The functional PC1-NNT interaction is also not fully validated. Even though the authors attempt to point to NNT deletion as the sole difference between N and J models, there could be differences in SNPs, epifactors, etc. Thus, this interaction is not conclusively validated. The key experiment is to show that deleting NNT abolishes the PC1-CTT rescue in the "N" model?

Thank you for your question and suggestions. We have added new data to the manuscript that supports the functional CTT/NNT interaction:

1 – In addition to the previous CTT-dependent increase in NNT enzymatic activity observed in mouse renal tissue, we now recapitulate these findings in an *in vitro* tissue culture system. We show that transfection of the 2HA-PC1-CTT construct increases NNT enzymatic activity in a mouse tubular cell line. Moreover, we show that the CTT capacity to increase NNT enzymatic activity is dependent upon the presence of its putative mitochondrial-targeting sequence. In fact, no changes in NNT enzymatic activity are observed after transfection with the same 2HA-PC1-CTT construct that lacks this 20-aa mitochondrial targeting sequence (revised Fig. 7 g-j).

2 – The CTT-dependent changes in NADH/NAD⁺ and NADPH/NADP⁺ previously observed in 16-week mice are present and more pronounced in pre-cystic 10-week mice, further supporting functional changes in NNT activity.

3 – In addition to the F1 NJ; *Pkhd1-Cre*; *Pkd1-KO* +/- CTT mice, we have now generated an F2 progeny by crossing F1x F1 NJ; *Pkd1^{f/+}*; *Pkhd1-Cre* +/- CTT mice. These F2 littermates were allocated to different cohorts based exclusively on *Nnt* WT or mutant homozygosity. F2 NN mice that expressed CTT recapitulated the phenotype suppression observed in F1 NJ CTT-expressing mice. F2 JJ littermates did not show any CTT-dependent changes in phenotype. Considering F2 NN and F2 JJ mice were littermates and selected solely based on their NNT status, this observation provides strong support for the functional relevance of the CTT/NNT interaction.

Additionally, as we pointed out in the original version of the manuscript, the *Crb1 rd8* alleles are the only other known loci with potential phenotypic relevance that vary between "N" and "J" mice¹⁸. Our analysis of genomic DNA from N-*Pkd1-KO* +/- CTT mice excludes the possibility

that skewed distributions of either of these mutant alleles could account for the observed phenotypic differences between the groups. Additionally, these observations suggest that any other potential independently assorted N versus J allelic variants are unlikely to be responsible for the observed phenotype differences.

While deleting NNT in the N mice could potentially provide some additional confirmation, these studies would be extremely time-consuming and require many months of breeding. We feel strongly that the extensive new data that we have added in response to the reviewer's concern provide substantial additional support for the importance of this functional and physical interaction. Of note, a second group has recently confirmed our findings by also identifying NNT as a PC1 binding partner (unpublished data, pre-print available at <https://www.biorxiv.org/content/10.1101/2022.04.08.487705v1.full.pdf>).

7) The metabolomics (Figure 5) and ADPH/NADP⁺ and NADH/NAD⁺ experiments (Figure 6) were done at cystic timepoints. The authors should have assessed them at a precystic timepoint. The differences could simply arise due to the choice of samples used for analysis. It's also very important that the cystic phenotype (kidney weights, BUN, creatinine) of the samples used are shown. Were the samples representative of the cohort phenotype?

Thank you for these important questions and comments. We agree that the metabolomics done at the 16-week timepoint, when cystic phenotype is fully developed, is useful in characterizing the metabolic phenotype of *Pkd1-KO* vs *Pkd1-KO+CTT* but is less useful in determining direct mechanistic consequences of CTT expression. The metabolites found in this scenario could arise as direct consequences of the CTT and NNT interaction but could also reflect the different degrees of cystic disease. Of note, the cystic phenotype of the animals used in these metabolomic studies were representative of the cohort phenotype, as shown in revised Supplementary Fig. 8c.

In light of the reviewer's very valuable suggestion, we repeated the metabolomics analysis at the pre-cystic 10-week timepoint, which showed significant changes in metabolism in N-*Pkd1-KO* mice that did or did not express the CTT. These findings are now shown in revised figure 6 and have been discussed extensively in the manuscript. Of note, the changes in NADH/NAD⁺ and NADPH/NADP⁺ are more pronounced at this early time point, suggesting that the secondary metabolic effects of cyst progression at later stages of the disease may actually be diluting these findings at the 16-week time point. Furthermore, the observed 5-fold change in ATP levels at the early time point is consistent with CTT exerting effects on mitochondrial function.

8) What is the relevance of reduced NNT expression in human samples? You may be comparing different cell types between ADPKD and healthy tissues. Did the authors find reduced NNT in mouse models? IF data should be shown also.

The reviewer raises an important point. We acknowledge the limitations of comparing surgically harvested renal tissues obtained from ADPKD patients who manifest a clear cystic phenotype with tissue obtained from biopsies of healthy controls. Because of these limitations, we have not furthered pursued efforts to understand the origins of the observed differences in NNT protein levels in human material. We have shown that NNT protein levels are indeed lower in 16-week N-*Pkd1-KO* mice (revised Fig. 5f,g), by which time the cystic phenotype is established. At the 10-week timepoint, prior to cystogenesis, there was no significant difference in NNT protein level (revised Fig. 7c,d). This makes it difficult to

determine whether changes in NNT protein levels arise as direct consequences of the CTT and NNT interaction or if these changes are secondary to other pathways that are perturbed as the cystic disease progresses. It is worth noting that, due to strong background staining in immunofluorescence, we chose to perform immunohistochemistry for NNT on mouse tissue, shown in revised Supplementary Fig. 4. These findings are discussed in the manuscript.

Reviewer #3 (Remarks to the Author):

This is an interesting study addressing a very important translational question. Various lines of evidence suggest that mitochondrial function and metabolism in kidney tubules are altered in ADPKD, but mechanistic explanations were lacking. The authors show that overexpression of a fragment of PKD1 (CTT) offers significant protection in a mouse model of ADPKD. Moreover, they provide evidence that CTT interacts with a critical mitochondrial enzyme (NNT) and affects its function, and that the beneficial effect of CTT is lost in mouse lacking this enzyme ("J" type mice). These are intriguing and potentially important observations, which could open the way to new therapeutic strategies. The manuscript is clearly written and the experimental approach is elegant and impressive.

However, the major conclusion of the authors – that NNT mediates the protective effect - essentially rests on the lack of response in "J" type mice, and while this is certainly plausible, it is difficult to be absolutely certain that they are not resistant for another, as yet undiscovered reason. In this alternative scenario, changes in mitochondrial NNT function (and expression of other mitochondrial enzymes/complexes) in response to CTT might be consequential rather than causal. Moreover, another issue is that NNT expression seems to be predominantly in the distal tubular segments, but some of the metabolic changes observed in response to CTT appear to be more pertinent to the proximal tubule (PT). Resolving these issues would further strengthen this exciting study.

I have the following comments:

1) As mentioned already, the major mechanistic issue is whether the protective effects of CTT are specifically mediated via NNT. This could in principle be resolved through depletion of the enzyme in ADPKD mice. Alternatively, the authors performed *in vitro* experiments with cells overexpressing CTT. Could they demonstrate in the cell model that CTT directly alters mitochondrial function, and that this is dependent on NNT? The mechanistic importance of demonstrating this is underlined by the fact that the authors were not able to show mitochondrial localization of CTT in kidney tissue by antibody staining, due to technical issues.

Thank you for your questions and suggestions. We agree that creating the mouse model that the reviewer proposed could provide further mechanistic confirmation. Considering, however, the extremely prolonged timeline that this experiment would entail, we chose to pursue the reviewer's alternative suggestion of generating an *in vitro* cell model to assess CTT-dependent changes in mitochondrial function. We generated an immortalized mouse tubular cell line and now show that transfection of the 2HA-PC1-CTT construct increases NNT enzymatic activity, consistent with the previous CTT-dependent increase in NNT enzymatic activity that we observed in mouse renal tissue. Moreover, we show that the CTT

capacity to increase NNT enzymatic activity is dependent upon the presence of its putative mitochondrial-targeting sequence. In fact, no changes in NNT enzymatic activity are observed after transfection with the same 2HA-PC1-CTT construct that lacks this 20-aa mitochondrial targeting sequence, revealing the functional specificity of this interaction (revised Fig. 7g-j). We then moved on to measuring relative NADH and NAD⁺ levels in these mitochondrial fractions by employing the NAD/NADH Quantitation Kit (catalog# MAK037, Sigma-Aldrich). The EV transfected cells revealed an average mitochondrial NADH/NAD⁺ ratio of ~0.3. This is within range of the mitochondrial NADH/NAD⁺ ratios described in the literature (~0.1-0.2)¹ and significantly out of range when compared to the expected NADH/NAD⁺ in the cytosolic fraction (~0.001-0.01), suggesting high specificity of our mitochondrial fraction isolation. Furthermore, the 2HA-PC1-CTT mitochondrial fractions revealed an average 6-fold increase in the NADH/NAD⁺ ratio when compared to the EV-transfected controls (Supplementary Fig. 10c). These findings suggest that the pronounced redox changes observed in 10-week pre-cystic bulk kidney tissue are likely a product of changes in mitochondrial redox secondary to CTT expression. Furthermore, the 5-fold change in ATP levels at this 10-week early metabolomic timepoint is consistent with CTT exerting effects on mitochondrial function.

Finally, we believe that the generation of the F2 *Pkhd1-Cre;Pkd1-KO +/-CTT* progeny, obtained by crossing F1xF1 NJ;*Pkd1^{fl/+};Pkhd1-Cre +/- CTT* mice, which is described in this revised manuscript, provides additional evidence of the importance of the CTT/NNT interaction in terms of disease progression. These F2 littermates were allocated to different cohorts based exclusively on *Nnt* WT or mutant homozygosity. F2 NN mice that expressed CTT recapitulated the phenotype suppression observed in F1 NJ CTT-expressing mice. F2 JJ littermates did not show any CTT-dependent changes in phenotype. Considering F2 NN and F2 JJ mice were littermates and selected solely based on their NNT status, these observations provide substantial data to support the functional relevance of the CTT/NNT interaction.

Of note, a second group has recently confirmed our findings by also identifying NNT as a PC1 binding partner and elucidating mild NNT-dependent phenotypes in the absence of CTT in PKD models (unpublished data, pre-print available at <https://www.biorxiv.org/content/10.1101/2022.04.08.487705v1.full.pdf>).

2)The authors state in suppl fig 3 that NNT is mostly expressed in distal tubular segments. However, it is difficult from the images provided to clearly appreciate this. More zoomed in images with tubular segment and mitochondrial markers would be helpful. What is known about NNT expression along the mouse nephron from gene expression databases? Assuming that the expression is more distal, many of the impressive effects of CTT (e.g. suppression of cyst growth) seem to be more widespread. Moreover, when the authors perform metabolic studies on whole kidney tissue (e.g. metabolomic screening, western blotting for mitochondrial enzymes), the readouts are likely to be dominated by PTs. For example, the authors highlight changes in the urea cycle – to the best of my knowledge this is occurring in the PT. This raises the question as to whether the metabolic changes observed in CTT overexpressing mice are mediated via NNT, or are rather occurring downstream to some other protective effect(s). This point is further reinforced by the beneficial effect of CTT in a collecting duct specific model of ADPKD. Following on the previous point, the authors could strengthen the notion that metabolic changes are

happening specifically in the NNT expressing tubular segments by performing antibody staining to assess mitochondrial mass and complex expression.

Thank you for your questions and suggestions. Higher quality and more zoomed-in images for NNT immunohistochemistry have been provided in revised Supplementary Fig. 4.

Unfortunately, immunofluorescence co-staining of NNT, although attempted, was limited by the high tissue background due to the use of a mouse-derived anti-NNT antibody.

Consequently, we needed to use histochemical staining and thus double labeling was not possible. We have added to the manuscript that the NNT distribution we observe in mice

(revised Supplementary Fig. 4) resembles that from both the Human Protein Atlas¹⁹ and the Rat Kidney Tubule Expression Atlas²⁰ (<https://esbl.nhlbi.nih.gov/KTEA/>).

While the metabolomics done at the 16-week timepoint, when cystic phenotype is fully developed, is useful in characterizing the metabolic phenotype of *Pkd1-KO* vs *Pkd1-KO+CTT*, we believe that it is less useful in determining proximate mechanistic consequences of CTT expression. The metabolites found in this scenario could arise as direct consequences of the CTT and NNT interaction but could also reflect the different degrees of cystic disease. Taking this into consideration, we repeated the metabolomics analysis at the pre-cystic 10-week timepoint, which showed significant changes in metabolism in *N-Pkd1-KO* mice that did or did not express the CTT. These findings are now shown in revised figure 6 and have been discussed extensively in the manuscript. Of note, the most impressive alteration found was the 5-fold increase in ATP levels in CTT-expressing mice which, in concert with the increase in levels of CIV and ATP-synthase and the late reduction in lactate levels, support the interpretation that CTT expression induces a profound shift towards normal metabolism in which oxidative phosphorylation serves as the predominant source of ATP generation. We acknowledge, however, that further experiments will be required to assess fully whether the observed metabolic changes do indeed reflect increased oxidative phosphorylation activity. Of note, the NNT-rich DCT segment is among the most mitochondria-rich segments of the kidney²¹, and the kidney altogether is the organ with the second highest mitochondrial content and oxygen consumption after the heart, which adds further potential mechanistic weight to the observed elevation of ATP levels secondary to the mitochondrial NNT-CTT interaction²².

3) Fig 3: Despite clearly developing cysts, the “J” type PKD1 KO mice do not appear to develop increases in BUN/creatinine compared to controls, as seen with the “N” type – do the authors have an explanation for this? Does it suggest that there may be baseline strain differences in response to PKD1 depletion?

Thank you for this question. Yes, we believe there are baseline strain differences in response to *Pkd1* depletion. Although our work is mostly focused on CTT effects in ADPKD, we have added Supplementary Fig 11 in which we re-plot *N-Pkd1-KO* vs *J-Pkd1-KO* data and show that serum creatinine levels were increased in *N-Pkd1-KO* mice. We now discuss this finding extensively in the “Discussion” section of the manuscript and believe that it provides interesting insight into the mechanism underlying CTT-dependent phenotype suppression. In summary, we propose that NNT-dependent ROS detoxification could allow for further proliferation of cystic tissue, similar to that observed in several cancer models¹⁴⁻¹⁶. These findings are supported by a previous study that has shown that disease progression in C57BL/6J was less severe than in the BC or 129 mice, which are expected to be NNT-expressing¹⁷. As mentioned in the manuscript, it is critical to point out that we do not

believe that NNT plays a direct role in cystogenic pathways. We believe, instead, that it acts as a disease modifier. Moreover, NNT itself is not a suppressor but, when influenced by CTT, it can exert suppressive effects. This finding suggests that NNT function needs to be increased by CTT and perhaps its functional properties modified even further (i.e., potential reversal of the direction of catalyzed enzymatic reaction in this pathological condition) as we extensively discuss in the “Discussion section” of this manuscript.

4) Fig 3: The authors use Ki67 staining to assess cell proliferation, but it has been argued by others that this might not be an accurate readout in tubular cells (PMID: 29632300). Moreover, it would again be helpful here to identify tubular segments.

Thank you for your comments and suggestions. We agree that the use of Ki67 staining to assess proliferation may have some limitations, but we believe it has been well-accepted as a marker in the PKD field in general. Many recent publications assess renal epithelial proliferation in ADPKD models using Ki67 staining^{4,5,23-25}.

Furthermore, we attempted staining with LTL and DBA lectins to identify proximal and distal segments, however performing antigen retrieval is necessary for Ki67 staining, and both of these lectins (which produce high-quality fluorescent signals in the absence of antigen retrieval) did not work appropriately in the presence of antigen retrieval. In this scenario, we believe that co-staining performed with the pan-tubular marker anti-Na,K-ATPase α subunit was appropriate, considering the *Pax8^{rtTa}* is a pan-tubular Cre promoter and therefore one can appreciate changes in proliferation in many different tubular segments.

5) Fig 6: Why is the redox state of both NADH and NADPH shifted towards reduced? Wouldn't opposite changes in each be expected if NNT activity is increased? Did the authors observe a change in the total pool sizes of NADH and NADPH? In the preceding figure, the authors show a relatively low expression of CI in the kidney (as demonstrated previously by others, consistent with the notion that the majority of electron flux is probably via CII). If respiratory chain activity and electron flux through CII are increased in CTT overexpressing animals, might this cause an increase in reverse electron flow through CI, pushing the NADH pool towards a reduced state independent of NNT?

Thank you for this interesting idea. It could be possible that the increase in the NADH pool, as the reviewer mentions, might be NNT-independent. We point out in the manuscript that while NAD(P)(H) measurements provide indirect insights into NNT function, the NAD(P)(H) pathways are extremely redundant (i.e., isocitrate dehydrogenase and malic enzyme have a relative contribution of 70% and 8%, respectively, to NADPH regeneration, while NNT is responsible for only 22%²) and that further studies will be necessary to fully elucidate the complete mechanism underlying NNT-dependent CTT suppression of cystic phenotype. Even so, the data supporting the relative increase in NADH and NADPH are extremely robust: we repeated the metabolomics analysis at the pre-cystic 10-week timepoint, which showed that the changes in NADH/NAD⁺ and NADPH/NADP⁺ are even more pronounced at this early time point. These observations suggest that the secondary metabolic effects of cyst progression at later stages of the disease may actually be diluting these findings at the later time point. Furthermore, *in vitro* findings from the renal epithelial cell system we generated (Supplementary Fig. 10) support the interpretation that the pronounced redox changes observed in 10-week pre-cystic bulk kidney tissue are a product of changes in mitochondrial redox secondary to CTT expression.

6) In their conclusions the authors state: "...these data support the interpretation that the CTT rescue model exhibits a profound shift towards oxidative phosphorylation as the predominant source of ATP generation". While the data are indeed consistent with this effect, the authors do not provide direct functional evidence to support the statement. As mentioned previously, this could be addressed with in vitro metabolic experiments (e.g. cells or isolated tubules). Or if not possible, the authors should at least clearly acknowledge this limitation.

Thank you for your questions and comments. As described in the responses to your comments 1 and 2 we believe we have provided more evidence supporting this shift in metabolism. We acknowledge in the discussion section, however, that this interpretation of the data is not 100% conclusive, with the following sentence: "Further experiments will be required to assess fully whether the observed metabolic changes do indeed reflect increased oxidative phosphorylation activity".

Minor:

7) It could be helpful for readers to better understand the background to the study if the authors included a schematic of PKD1 structure and relevant fragments.

Thank you for the thoughtful suggestion. This schematic has been provided (revised Fig. 8).

8) Fig 2: The authors should include TOM20 images to clearly demonstrate the mitochondrial location of CTT.

Thank you for the thoughtful suggestion. While CTT and TOMM20 IF staining could not be performed simultaneously since both primary antibodies are derived in rabbits (#EJH002, Kerafast and #NBP1-81556, Novus Biologicals, respectively), we have added revised Supplementary Fig.3 showing clear mitochondrial colocalization between TOMM20 and NNT (quantified in revised Fig. 2f). Considering the same NNT antibody (#459170, Invitrogen) was used in revised Fig. 2e, we believe this is strong evidence to support mitochondrial localization of CTT.

9) Abstract: The authors state that CTT overexpression "dramatically suppresses cystic phenotype". While the effect was clearly significant, it seems more of a partial effect? Please consider rewriting.

Thank you for the thoughtful suggestion. We have rewritten this part of the abstract.

Reviewer #4 (Remarks to the Author):

Summary: Autosomal dominant polycystic kidney disease (ADPKD) is the most prevalent and potentially lethal monogenic disorder. Mutations in the PKD1 gene, encoding the PC1 protein, account for most cases. Post-translation proteolytic processing of PC1 generates a C-terminal fragment that is translocated to the mitochondria. The peptide comprising the 200 C-terminal amino acids suppresses the cystic phenotype in a PKD1 knockout mouse model of ADPKD. Proteomics data demonstrate this 200 aa fragment physically interacts with the mitochondrial enzyme nicotinamide nucleotide transhydrogenase (NNT). This interaction also impacts mitochondrial stoichiometry for TOMM20, complex V and cytochrome c oxidase. The C-terminal fragment inter Metabolomics data suggests that

expression of this peptide in the PKD1 KO animals normalizes tissue levels of methionine, lactate, asparagine and glutamate among others. Data also suggest that the 200 C-terminal fragment interacting with NNT reduces the activity of NNT and diminishes the NADPH to NADP+ and the NADH to NAD+ ratio. Taken together the data suggest that the interaction of the 200 C-terminal fragment of PC1 interacts with NNT modifies the disease course and is not directly causally associated with the reduction in cystogenic pathways. It is encouraging that there is a potential for gene delivery of a smallish protein to mitigate the progressive course of a profound and prevalent disease.

Comments: Very elegant work to address the research question by developing competing animal models to subtract and add back in the PC1 protein and domains. Solid proteomics, metabolomics, immunofluorescence and immunoblot work.

(1) Previous pathways analysis of proteomics work demonstrated associated of the 200aa C-terminal PC1 protein to the mitochondrial matrix. Crude mitochondrial preparations containing the mitochondrial matrix and mitochondrial-associated endoplasmic reticulum membranes was used for identifying protein-protein interaction studies. These studies rely on DTSSP, a chemical crosslinker, to stabilize and enrich protein-protein interactions and specific binding partner data. The data demonstrate a specific enrichment in the interaction with NNT. Three missed opportunities exist here- (1) site specific identification of protein-protein interactions- DTSSP is a reducible chemical crosslinker. Proteolytic digestion of the DTSSP crosslinked sample without reduction & alkylation may present with an opportunity to identify crosslinked peptides. This method would require enrichment of digested peptides using a PD-10 type column and collection of data on +4/+5/+6 charge state ions and then analysis of the data by one of several available apps such as reviewed in Slavin M et al Anal. Chem. 2020, 92, 15899–15907. This might strengthen the information for the direct interactions of the C-terminus of PC1 with NNT or other mitochondrial matrix proteins. (2) why didn't the authors refine the volcano plot with PPI using Stringdb or other tools to understand how NNT might be interaction with other proteins in their samples. (3) What is the pathways analysis for these interactors.

Thank you for your question and very interesting suggestions. We agree that the proposed analysis has the potential to be quite informative regarding site-specific aspects of the observed interactions. Undertaking this line of analysis would require extensive and lengthy experimental effort and refinement. It is worth noting that we share the reviewer's interest in understanding the nature of the NNT/CTT interaction and we are currently engaged in on-going collaborating collaborative studies with structural biologists that are designed to identify the protein interaction domains that are involved. We hope that the reviewer will agree with our strong feeling that these structural studies are beyond the scope of the present manuscript. We did not refine our analysis with pathway analysis because we only identified 13 potential interactors meeting both criteria of P value <0.05 and fold change >2 (all are identified in Supplementary table 1). Since small pathways can exhibit false positive associations and we found ourselves very close to the minimum of 10 genes²⁶ for pathway analyses, we decided to proceed with the evaluation of NNT, which was far-and-away the most enriched interactor. Of note, we point out the relevance of GANAB in the manuscript.

(2) The authors use metabolomics-based profiling of renal tissue to understand how the 200 C-terminal aa of PC1 impact renal parenchymal metabolism and identified dysregulated metabolomic features including three amino acids and lactic acid. While interesting I am

not sure how this adds to the story. How does dysregulation of these amino acids tie in with the observed levels of reduced NADPH or NADH? Is this logical or a true-true but unrelated observation.

Thank you for this interesting question. We did take into consideration that these 16-week metabolomics findings could constitute true-true but unrelated observations, even more so considering that metabolites found in this scenario, while important in terms of phenotype characterization, could arise as direct consequences of the CTT and NNT interaction but could also just reflect the different degrees of cystic disease. Taking this into consideration, we repeated the metabolomics analysis at the pre-cystic 10-week timepoint, which showed significant changes in metabolism in *N-Pkd1*-KO mice that did or did not express the CTT. These findings are now shown in revised figure 6 and have been discussed extensively in the manuscript. Of note, the changes in NADH/NAD⁺ and NADPH/NADP⁺ are more pronounced at this early time point, suggesting that the secondary metabolic effects of cyst progression at later stages of the disease may actually be diluting these differences at the later time point. Furthermore, the 5-fold change in ATP levels at the early timepoint is consistent with the relative increase in NADH and consistent with the increased level of ATP-synthase detected, suggesting that these pre-cystic findings are indeed more relevant from a mechanistic perspective.

(3) Supplemental information on proteomics experiments is poor. Information on crosslinking mass spectrometry studies is completely absent. Given that the method for metabolomics is very detailed it would be essential to provide the same level of detail for the proteomics experiments. (Especially on how the crosslinking studies were conducted and MS data analyzed).

Thank you for the question and suggestion. The details regarding the crosslinking studies are provided in the “Immunoprecipitation from crude mitochondria fractions prepared from *Pkd1*^{F/H}-BAC mice” section of methods. Further details regarding the proteomics experiments have been added to the methods section, with the help and support of Dr. TuKiet Lam, Ph.D. Director, Keck MS & Proteomics Resource, since this experiment was conducted in association with the core service that he directs.

BIBLIOGRAPHY

1. Yang, Y. & Sauve, A.A. NAD(+) metabolism: Bioenergetics, signaling and manipulation for therapy. *Biochim Biophys Acta* **1864**, 1787-1800 (2016).
2. Nickel, A.G., von Hardenberg, A., Hohl, M., Loffler, J.R., Kohlhaas, M., Becker, J. *et al.* Reversal of Mitochondrial Transhydrogenase Causes Oxidative Stress in Heart Failure. *Cell Metab* **22**, 472-84 (2015).
3. Fedeles, S.V., Tian, X., Gallagher, A.R., Mitobe, M., Nishio, S., Lee, S.H. *et al.* A genetic interaction network of five genes for human polycystic kidney and liver diseases defines polycystin-1 as the central determinant of cyst formation. *Nat Genet* **43**, 639-47 (2011).
4. Ma, M., Tian, X., Igarashi, P., Pazour, G.J. & Somlo, S. Loss of cilia suppresses cyst growth in genetic models of autosomal dominant polycystic kidney disease. *Nat Genet* **45**, 1004-12 (2013).

5. Dong, K., Zhang, C., Tian, X., Coman, D., Hyder, F., Ma, M. *et al.* Renal plasticity revealed through reversal of polycystic kidney disease in mice. *Nat Genet* (2021).
6. Cassini, M.F., Kakade, V.R., Kurtz, E., Sulkowski, P., Glazer, P., Torres, R. *et al.* Mcp1 Promotes Macrophage-Dependent Cyst Expansion in Autosomal Dominant Polycystic Kidney Disease. *J Am Soc Nephrol* **29**, 2471-2481 (2018).
7. Ma, M., Legue, E., Tian, X., Somlo, S. & Liem, K.F., Jr. Cell-Autonomous Hedgehog Signaling Is Not Required for Cyst Formation in Autosomal Dominant Polycystic Kidney Disease. *J Am Soc Nephrol* **30**, 2103-2111 (2019).
8. Harris, P.C. & Rossetti, S. Determinants of renal disease variability in ADPKD. *Adv Chronic Kidney Dis* **17**, 131-9 (2010).
9. Lanktree, M.B., Guiard, E., Li, W., Akbari, P., Haghighi, A., Iliuta, I.A. *et al.* Intrafamilial Variability of ADPKD. *Kidney Int Rep* **4**, 995-1003 (2019).
10. Duplus-Bottin, H., Spichty, M., Triqueneaux, G., Place, C., Mangeot, P.E., Ohlmann, T. *et al.* A single-chain and fast-responding light-inducible Cre recombinase as a novel optogenetic switch. *Elife* **10**(2021).
11. Luippold, G., Pech, B., Schneider, S., Osswald, H. & Muhlbauer, B. Age dependency of renal function in CD-1 mice. *Am J Physiol Renal Physiol* **282**, F886-90 (2002).
12. Raman, A., Reif, G.A., Dai, Y., Khanna, A., Li, X., Astleford, L. *et al.* Integrin-Linked Kinase Signaling Promotes Cyst Growth and Fibrosis in Polycystic Kidney Disease. *J Am Soc Nephrol* **28**, 2708-2719 (2017).
13. Piontek, K., Menezes, L.F., Garcia-Gonzalez, M.A., Huso, D.L. & Germino, G.G. A critical developmental switch defines the kinetics of kidney cyst formation after loss of Pkd1. *Nat Med* **13**, 1490-5 (2007).
14. Li, S., Zhuang, Z., Wu, T., Lin, J.C., Liu, Z.X., Zhou, L.F. *et al.* Nicotinamide nucleotide transhydrogenase-mediated redox homeostasis promotes tumor growth and metastasis in gastric cancer. *Redox Biol* **18**, 246-255 (2018).
15. Chortis, V., Taylor, A.E., Doig, C.L., Walsh, M.D., Meimaridou, E., Jenkinson, C. *et al.* Nicotinamide Nucleotide Transhydrogenase as a Novel Treatment Target in Adrenocortical Carcinoma. *Endocrinology* **159**, 2836-2849 (2018).
16. Ho, H.Y., Lin, Y.T., Lin, G., Wu, P.R. & Cheng, M.L. Nicotinamide nucleotide transhydrogenase (NNT) deficiency dysregulates mitochondrial retrograde signaling and impedes proliferation. *Redox Biol* **12**, 916-928 (2017).
17. Arroyo, J., Escobar-Zarate, D., Wells, H.H., Constans, M.M., Thao, K., Smith, J.M. *et al.* The genetic background significantly impacts the severity of kidney cystic disease in the Pkd1(RC/RC) mouse model of autosomal dominant polycystic kidney disease. *Kidney Int* **99**, 1392-1407 (2021).
18. Simon, M.M., Greenaway, S., White, J.K., Fuchs, H., Gailus-Durner, V., Wells, S. *et al.* A comparative phenotypic and genomic analysis of C57BL/6J and C57BL/6N mouse strains. *Genome Biol* **14**, R82 (2013).
19. Uhlen, M., Fagerberg, L., Hallstrom, B.M., Lindskog, C., Oksvold, P., Mardinoglu, A. *et al.* Proteomics. Tissue-based map of the human proteome. *Science* **347**, 1260419 (2015).
20. Limbutara, K., Chou, C.L. & Knepper, M.A. Quantitative Proteomics of All 14 Renal Tubule Segments in Rat. *J Am Soc Nephrol* **31**, 1255-1266 (2020).
21. Subramanya, A.R. & Ellison, D.H. Distal convoluted tubule. *Clin J Am Soc Nephrol* **9**, 2147-63 (2014).

22. Bhargava, P. & Schnellmann, R.G. Mitochondrial energetics in the kidney. *Nat Rev Nephrol* **13**, 629-646 (2017).
23. Cabrita, I., Kraus, A., Scholz, J.K., Skoczynski, K., Schreiber, R., Kunzelmann, K. *et al.* Cyst growth in ADPKD is prevented by pharmacological and genetic inhibition of TMEM16A in vivo. *Nat Commun* **11**, 4320 (2020).
24. Zhang, X., Li, L.X., Ding, H., Torres, V.E., Yu, C. & Li, X. Ferroptosis Promotes Cyst Growth in Autosomal Dominant Polycystic Kidney Disease Mouse Models. *J Am Soc Nephrol* **32**, 2759-2776 (2021).
25. Ding, H., Li, L.X., Harris, P.C., Yang, J. & Li, X. Extracellular vesicles and exosomes generated from cystic renal epithelial cells promote cyst growth in autosomal dominant polycystic kidney disease. *Nat Commun* **12**, 4548 (2021).
26. Ramanan, V.K., Shen, L., Moore, J.H. & Saykin, A.J. Pathway analysis of genomic data: concepts, methods, and prospects for future development. *Trends Genet* **28**, 323-32 (2012).

REVIEWERS' COMMENTS

Reviewer #1 (Remarks to the Author):

The authors have addressed all of my comments. I congratulate them on this interesting and important contribution.

Reviewer #2 (Remarks to the Author):

The authors have been responsive, addressed many issues by performing additional experiments, and the manuscript is improved. I support the manuscript for publication.

Reviewer #3 (Remarks to the Author):

The authors have provided detailed responses to my comments and strengthened the manuscript substantially with considerable amounts of new data. The new in vitro experiments provide further support for the proposed mechanism. The critical experiment of depleting NNT in vivo is still outstanding, but due to the timelines required to perform this it is probably best addressed in a follow up study. I have no further comments.

Reviewer #4 (Remarks to the Author):

Onuchic L et al

The C-terminal tail of polycystin-1 suppresses cystic disease in a mitochondrial enzyme-dependent fashion.

Summary: Autosomal dominant polycystic kidney disease (ADPKD) is the most prevalent and potentially lethal monogenic disorder. Mutations in the PKD1 gene, encoding the PC1 protein, account for most cases. Post-translation proteolytic processing of PC1 generates a C-terminal fragment that is translocated to the mitochondria. The peptide comprising the 200 C-terminal amino acids suppresses the cystic phenotype in a PKD1 knockout mouse model of ADPKD. Proteomics data demonstrate this 200 aa fragment physically interacts with the mitochondrial enzyme nicotinamide nucleotide transhydrogenase (NNT). This interaction also impacts mitochondrial stoichiometry for TOMM20, complex V and cytochrome c oxidase. The C-terminal fragment inter Metabolomics data suggests that expression of this peptide in the PKD1 KO animals normalizes tissue levels of methionine, lactate, asparagine and glutamate among others. Data also suggest that the 200 C-terminal fragment interacting with NNT reduces the activity of NNT and diminishes the NADPH to NADP+ and the NADH to NAD+ ratio. Taken together the data suggest that the interaction of the 200 C-terminal fragment of PC1 interacts with NNT modifies the disease course and is not directly causally associated with the reduction in cystogenic pathways. It is encouraging that there is a potential for gene delivery of a smallish protein to mitigate the progressive course of a profound and prevalent disease.

Three comments were provided in the primary review of this manuscript and the authors have address all three to some extent. Regarding comment one, the reviewer appreciates the caution the authors took in consideration of small numbers of target interacting proteins and the potential to over interpret the significance in pathways analysis tools. Regarding comment two, the reviewer is satisfied with the authors revisions. Regarding comment three the reviewer very much appreciates their support of disseminating these methods used in their research.

No further comments or critiques are raised.